# Comparing SAM 2 and SAM 3 for Zero-Shot Segmentation of 3D Medical Data

**Satrajit Chakrabarty**[1]  (iD)        SATRAJIT.CHAKRABARTY@GEHEALTHCARE.COM

**Ravi Soni**[1]        RAVI.SONI@GEHEALTHCARE.COM

[1] *GE HealthCare, San Ramon, CA, USA*

**Editors:** Accepted for publication at MIDL 2026

## Abstract

Foundation models, such as the Segment Anything Model (SAM), have heightened interest in promptable zero-shot segmentation. Although these models perform strongly on natural images, their behavior on medical data remains insufficiently characterized. While SAM 2 has been widely adopted for annotation in 3D medical workflows, the recently released SAM 3 introduces a new architecture that may change how visual prompts are interpreted and propagated. Therefore, to assess whether SAM 3 can serve as an out-of-the-box replacement for SAM 2 for zero-shot segmentation of 3D medical data, we present the first controlled comparison of both models by evaluating SAM 3 in its Promptable Visual Segmentation (PVS) mode using a variety of prompting strategies. We benchmark on 16 public datasets (CT, MRI, Ultrasound, endoscopy) covering 54 anatomical structures, pathologies, and surgical instruments. We further quantify three failure modes: prompt-frame over-segmentation, over-propagation after object disappearance, and temporal retention of well-initialized predictions. Our results show that SAM 3 is consistently stronger under click prompting across modalities, with fewer prompt-frame over-segmentation failures and slower prediction retention decay compared to SAM 2. Under bounding-box and mask prompts, performance gaps narrow in few structures of CT/MR and the models trade off termination behavior, while SAM 3 remains stronger on ultrasound and endoscopy sequences. The overall results position SAM 3 as the superior default choice for most medical segmentation tasks, while clarifying when SAM 2 remains a preferable propagator.

**Keywords:** Foundation models, Segment Anything Model, Zero-shot segmentation, SAM 2, SAM 3.

## 1. Introduction

Foundation models for promptable segmentation have reshaped interactive medical image analysis. The Segment Anything Model (SAM) (Kirillov et al., 2023) introduced a general-purpose framework for zero-shot segmentation of 2D images using point, box, and mask prompts. SAM 2 (Ravi et al., 2024) extended this approach to videos and 3D-like sequences with a memory-based transformer for frame-to-frame propagation, enabling consistent segmentation across volumes and cine series. The most recent iteration, SAM 3 (Carion et al., 2025) introduces a unified Perception Encoder and a DETR (DEtection TRansformer)-style detector–tracker (Carion et al., 2020), and adds concept-level prompting modules for open-vocabulary segmentation.

Several medical variants of the SAM-family models have been proposed through domain-specific training, including supervised adaptation on curated medical corpora (e.g., MedSAM (Ma et al., 2024a), MedSAM2 (Ma et al., 2025)) and synthetic-data-driven training

that reduces reliance on real annotations (e.g., SynthFM (Sengupta et al., 2025), SynthFM-3D (Chakrabarty et al., 2026)). However, the original SAM-family models remain practically relevant in medical workflows when the goal is interactive, human-in-the-loop annotation or rapid dataset bootstrapping rather than fully automated clinical segmentation. In such cold-start settings involving new anatomies, new devices, or new sites, labeled training sets may be unavailable, and a promptable zero-shot model can produce a first-pass mask that a human corrects to efficiently generate training data for downstream supervised models.

In this context, an important open question is whether SAM 3 can serve as an out-of-the-box replacement for SAM 2 under purely visual prompting. SAM 3 supports two regimes: Promptable Concept Segmentation (PCS), which enables concept/text-conditioned outputs, and Promptable Visual Segmentation (PVS), which operates from purely visual prompts (Carion et al., 2025). Although PCS expands the model's scope, the architectural changes introduced in SAM 3 may also alter how visual prompts are interpreted and how masks are propagated over long medical sequences. To answer this question, we conduct a large-scale, controlled comparison of SAM 2 and SAM 3 across sixteen public datasets spanning CT, MRI, ultrasound, and endoscopy, covering 54 anatomical structures, pathologies, and surgical instruments. We evaluate SAM 3 in the PVS regime using only visual prompts and no concept prompts (i.e., no text or exemplar inputs), so that both models operate under matched prompting and propagation conditions. We benchmark single-click, multi-click, bounding-box, and mask prompts applied only to the first frame. Beyond prompt-frame and full-volume performance, we quantify three failure modes that are critical for interactive workflows: prompt-frame over-segmentation (poor initialization), temporal retention (forgetting), and over-propagation after object disappearance.

This study makes three main contributions:

- A unified, cross-modality evaluation framework for comparing SAM 2 and SAM 3 under identical visual prompts

- A comprehensive empirical characterization of prompt-frame and full-volume/sequence performance across sixteen datasets, spanning four modalities and 54 targets, under single-click, multi-click, bounding-box, and mask prompting.

- A cross-model failure-mode analysis that quantifies prompt-frame over-segmentation, temporal decay of prediction, and over-propagation, providing the first systematic evidence on when SAM 3 can serve as an out-of-the-box replacement for SAM 2 and when SAM 2 remains the more conservative propagator.

By isolating visual-prompt behaviour and conducting extensive cross-modality experiments, this work clarifies the complementary strengths of SAM 2 and SAM 3 and provides practical guidance for selecting between these models in clinical and research settings.

## 2. Methods

### 2.1. Comparison rationale and scope

The objective of this study is to compare SAM 2 and SAM 3 under controlled and identical prompting conditions for medical image segmentation in 3D volumes and medical video se-

quences. SAM 3 expands the SAM 2 family with both PVS and PCS (Carion et al., 2025). Because our goal is a controlled comparison under the same user interaction assumptions used in medical annotation workflows, we evaluate both models using only visual prompts (points, boxes, masks) and identical first-frame initialization followed by forward propagation. This scope isolates differences in visual prompt interpretation and propagation dynamics, and avoids introducing unmatched semantic inputs that would confound attribution. A key motivation for our controlled study is that it provides architectural and prompting insights that can inform future medically adapted variants of SAM 3, including design choices around PVS-style prompting and propagation behavior.

This scope also defines what constitutes a fair baseline: we study the zero-shot behavior of the released SAM-family models only on 3D medical data without any domain-specific training or task optimization. Accordingly, methods whose performance is intrinsically tied to medical training data or task-specific supervision are not included as baselines in our study, including 2D-only promptable models (e.g., MedSAM (Ma et al., 2024a)), medically fine-tuned SAM 2 derivatives (e.g., MedSAM2 (Ma et al., 2025)), and fully supervised task-trained pipelines (e.g., nnU-Net (Isensee et al., 2021)). Since a fair comparison to supervised baselines would also require medical fine-tuning of SAM 2/SAM 3 under matched data and protocols, we focus our analysis on stress-testing out-of-the-box prompting and propagation behavior. Further discussion of scope and comparability assumptions is provided in Appendix A.

## 2.2. Model Overview

SAM 2 (Ravi et al., 2024) is an encoder–decoder architecture built on the Hiera (Hierarchical Vision Transformer) backbone (Ryali et al., 2023). Its defining feature is a streaming memory mechanism designed for semi-supervised video object segmentation where a memory bank stores features and masks from past frames, and a memory-attention module aggregates these to enforce spatio-temporal consistency during propagation through a 3D volume or cine sequence. SAM 3 (Carion et al., 2025) uses a unified Perception Encoder shared by a DETR-style detector and a tracker. The detector follows the DETR paradigm with learnable object queries for localization/association (Carion et al., 2020), while the tracker inherits the SAM 2 transformer encoder-decoder for video segmentation and interactive refinement and retains a SAM 2-style propagation mechanism with a memory encoder and memory bank.

In this work, we evaluate SAM 3 in the PVS mode, so that SAM 3 operates as a visual promptable tracker and segmenter under the same interaction protocol as SAM 2. Importantly, in all our experiments, this mode is realized by inference-path selection where we use the official tracker-based visual-prompt interface released by the authors so that concept-conditioning modules are not exercised. Appendix B documents the exact inference interface we build upon.

Under our fixed visual-prompt protocol, cross-model differences are interpreted through (i) the learned visual representation (Hiera backbone versus the unified Perception Encoder) and (ii) the propagation dynamics induced by the two tracking/memory formulations, which together govern initialization quality, retention, and termination over long sequences. More

details on their differences, and a component-level summary linking these differences to different failure modes discussed in the paper, are provided in Appendix C.

## 2.3. Prompting Strategy

We evaluate three standard visual prompting strategies: (i) *click prompting*, using either a single positive click (1,0) or a mixed positive–negative configuration (1,2), where the positive click is placed near the centroid of the target and negative clicks are sampled from a dilated region around the structure; (ii) *bounding-box prompting*, where a tight axis-aligned box around the ground-truth structure in the first frame provides coarse geometric context; and (iii) *mask prompting*, where a binary ground-truth mask from the first frame in which the structure appears is supplied as the initial prompt. All prompts are provided only on the first frame. Thereafter, the models receive no iterative prompting or corrective interactions and propagate their predictions sequentially from the first to the last frame without forward–backward refinement, temporal smoothing, or post-processing. For click prompts, we avoid oracle-style interactive prompting (i.e., placing subsequent clicks using ground-truth error regions conditioned on a model's intermediate predictions) and instead use a fixed, model-independent initialization prompt so that SAM 2 and SAM 3 receive identical click inputs. Full details of the prompting protocol and a click-jitter robustness study are provided in Appendix D.

## 2.4. Datasets and Implementation Details

Table 1: Descriptions of the medical imaging datasets used for evaluation.

| Modality | Dataset | Anatomy | #Volumes / #Frames | #Classes |
|---|---|---|---|---|
| CT | AMOS | Abdominal | 200 / 26069 | 9 |
| | BTCV | Abdominal | 30 / 3779 | 13 |
| | FLARE22 | Abdominal | 50 / 4794 | 13 |
| | MSD Lung | Lung tumor | 63 / 17657 | 1 |
| | MSD Pancreas | Pancreas, tumor | 281 / 26719 | 2 |
| | MSD Spleen | Spleen | 41 / 3650 | 1 |
| | MSD Colon | Colon cancer | 126 / 13486 | 1 |
| | TotalSegmentator | Abdominal | 1113 / 304346 | 13 |
| MRI | ACDC | Cardiac | 149 / 1482 | 3 |
| | AMOS | Abdominal | 40 / 9455 | 9 |
| | MSD Heart | Cardiac | 20 / 2271 | 2 |
| | MSD Hippocampus | Hippocampus | 260 / 9270 | 2 |
| | TotalSegmentator | Abdominal | 1880 / 287217 | 13 |
| US | CAMUS | Cardiac | 500 / 9964 | 3 |
| | SegThy | Thyroid/vascular | 32 / 15820 | 5 |
| Endoscopy | CholecSeg8K | Cholecystectomy | 17 / 8080 | 12 |

We evaluate SAM 2 and SAM 3 on sixteen publicly available medical imaging datasets spanning four imaging modalities: 3D CT, 3D MRI, ultrasound (2D cine and 3D volumes), and endoscopy video (Table 1, Figure 1). All evaluated datasets are either 3D volumes or temporal sequences; we do not include any 2D datasets. Our data selection covers a broad spectrum of anatomical structures, pathological conditions, and clinical instruments across modalities, ensuring that the evaluation reflects the diversity encountered in real-world

clinical imaging workflows. The CT cohorts include multi-organ abdominal benchmarks (AMOS (Ji et al., 2022), BTCV (Landman et al., 2015), FLARE22 (Ma et al., 2024b), TotalSegmentator (Wasserthal et al., 2023)) together with oncologic and thoracic tasks from the MSD collection (lung tumors, pancreas and pancreatic tumors, spleen, and colon cancer) (Antonelli et al., 2022; Simpson et al., 2019). MRI coverage comes from AMOS22 (Ji

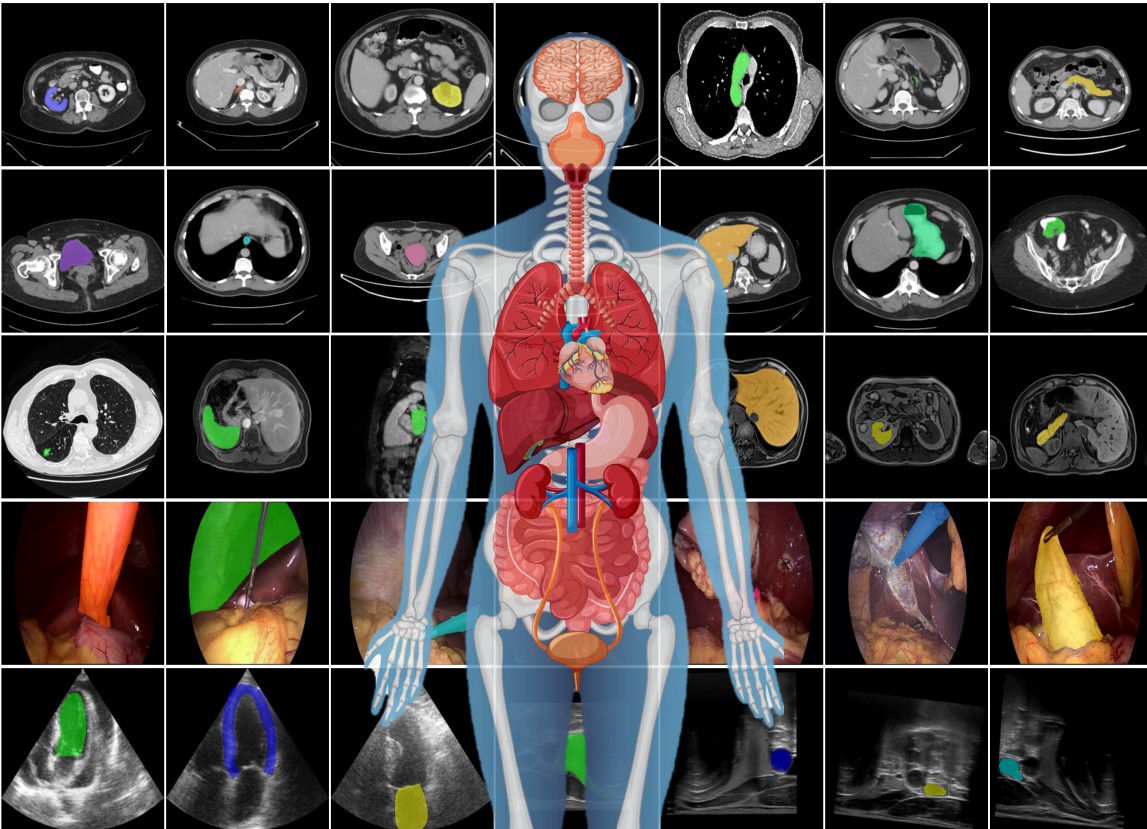

Figure 1: **Overview of the multi-modality benchmark dataset.** Representative images with ground-truth masks from the sixteen public datasets used in this study, illustrating variability in modality, anatomy, pathology, contrast, and acquisition. [Human illustration adapted from Vecteezy.com].

et al., 2022), ACDC (Bernard et al., 2018), MSD Task02 Heart and Task04 Hippocampus (Antonelli et al., 2022; Simpson et al., 2019), and the MRI subset of TotalSegmentator (D'Antonoli et al., 2024). Ultrasound is represented by cardiac cine sequences from CAMUS (Leclerc et al., 2019) and 3D thyroid ultrasound from SegThy (Krönke et al., 2022), while CholecSeg8K (Hong et al., 2020; Twinanda et al., 2016) provides endoscopy video frames with organ and instrument labels. Following prior work that treats 3D medical volumes as video-like slice sequences for SAM-style models, where each slice can be treated as a frame, we represent each 3D scan as an ordered sequence of 2D slices and process it sequentially (Dong et al., 2024; Ma et al., 2025). Segmentation accuracy is measured using the

Dice similarity coefficient (DSC). Statistical significance is assessed using paired Wilcoxon signed-rank tests on video/volume-level DSC, with significance defined at $\alpha = 0.05$. All preprocessing and checkpoint details are provided in Appendix E.

### 2.5. Failure Mode Analysis

To complement volume-level DSC, we quantify three failure modes at a *case* level, where *case* is a single target structure within a volume per dataset. All metrics are computed per case and summarized as distributions across all cases, stratified by prompting mode.

***1. Prompt-frame oversegmentation (flooding).*** To measure whether the model accurately resolves the target's spatial extent or "floods" into the background, we compute an *area ratio* on the prompt frame $t_0$. Let $M_{gt}^{(t_0)}$ and $M_{pred}^{(t_0)}$ denote the ground-truth and predicted binary masks at $t_0$, and $|\cdot|$ the foreground pixel count. For all cases with $|M_{gt}^{(t_0)}| > 0$ we define

$$R = \frac{|M_{pred}^{(t_0)}|}{|M_{gt}^{(t_0)}|}, \tag{1}$$

and analyze both the distribution of $R$ and the fraction of *severe flooding* events ($R > 2$).

***2. Temporal retention (forgetting).*** To measure how segmentation quality evolves across a volume/sequence while the object is present, we model the decay of Dice over the object's lifespan. For each case, we consider all frames where the ground-truth mask is non-empty and DSC is defined, re-index the frame IDs to a normalized time variable $\tau \in [0, 1]$, and fit a simple linear model, $DSC(\tau) \approx \alpha + \beta \tau$. The *normalized decay slope* $\beta$ serves as a retention score: values closer to zero indicate stable performance, whereas more negative $\beta$ correspond to faster forgetting. We compute $\beta$ for all cases as well as focus on a subset of cases with good initialization (prompt-frame $DSC \geq 0.7$).

***3. Over-propagation after object disappearance.*** To quantify how long a model continues to hallucinate a mask after the physical object has disappeared, we count the number of *over-propagated frames*. In our evaluation, each case is an ordered sequence/volume with frames indexed by $t$. Let the target ground-truth mask at frame $t$ be $M_{gt}^{(t)}$ and the model prediction mask be $M_{pred}^{(t)}$. Let $t_{\text{last}}$ denote the final frame where the target ground-truth mask is non-empty ($M_{gt}^{(t_{\text{last}})} \neq \emptyset$). The over-propagation length for a case is then

$$L = \#\{t > t_{\text{last}} \mid M_{gt}^{(t)} = \emptyset \ \wedge \ M_{pred}^{(t)} \text{ is non-empty}\},$$

i.e., the number of frames after $t_{\text{last}}$ where the target ground-truth mask is empty but the model continues to hallucinate and keeps incorrectly producing a non-empty prediction. Note that, if the target ground-truth mask persists through the final frame of the sequence/volume, then no post-$t_{\text{last}}$ frames exist and $L = 0$ by definition (e.g., the CA-MUS dataset). We summarize $L$ via boxen plots and empirical cumulative distribution functions, and report percentiles such as the 90th percentile ($P_{90}$), which indicates the over-propagation length below which 90% of cases fall.

## 3. Results

### 3.1. Prompt-Frame Accuracy

To isolate the effect of prompt interpretation, defined here as the model's ability to accurately resolve the spatial extent of the target structure on the initial frame based on user input, we measured segmentation performance on the prompt-frame only. While detailed numerical results for all 54 anatomical structures are provided in Appendix F (Table 4), Figure 2 summarizes two key aspects: (a) the distribution of the prediction-to-ground-truth area ratio $R$ for each prompt type on a log scale, and (b) the fraction of cases with severe over-segmentation ($R > 2$) stratified by object ground-truth size.

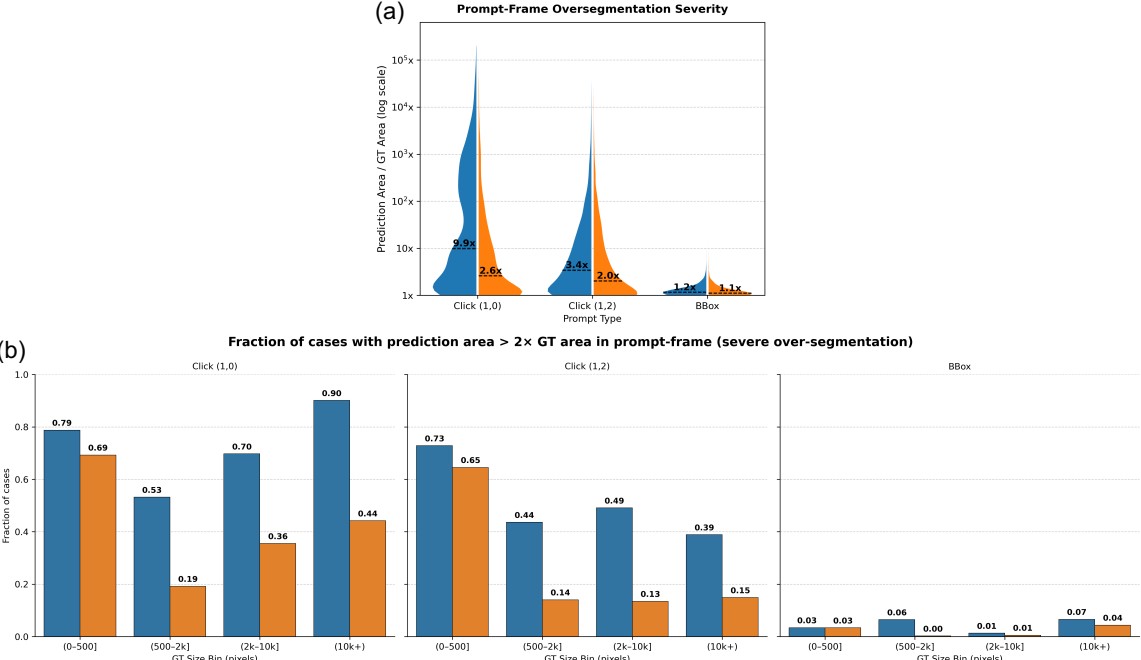

Figure 2: **Analysis of Prompt-Frame Over-segmentation.** **(a)** Distribution of the $log_{10}$ Area Ratio ($A_{pred}/A_{gt}$). **(b)** Proportion of severe oversegmentation failures ($A_{pred} > 2 \times A_{gt}$), stratified by ground truth object size. [Color: SAM 2, SAM 3]

Across all structures and prompt types, SAM 3 provides markedly stronger and more stable initialization than SAM 2. Under click prompting, SAM 2 exhibits a severe instability manifested as a heavy-tailed distribution in Figure 2a: its predicted masks are frequently $10\times$ to $10^5\times$ larger than the ground truth. Specifically, the median area ratio for single-click prompts is $9.9\times$ for SAM 2 versus $2.6\times$ for SAM 3; for multi-click prompts, the medians drop to $3.4\times$ and $2.0\times$, respectively, and for bounding boxes they are close to unity at $1.16\times$ (SAM 2) and $1.11\times$ (SAM 3). Thus, even under sparse clicks, SAM 3 keeps the predicted area much closer to the ground-truth support, whereas SAM 2 frequently floods large portions of the frame.

Figure 2b quantifies the frequency of the severe over-segmentations ($R > 2$). For single-click prompting on small targets ($< 500$ pixels), 79% of SAM 2 initializations are severely over-segmented, compared with 69% for SAM 3; for medium-sized structures (500–2k pixels), the gap widens to 53% vs. 19%, and even for the largest objects ($\geq 10k$ pixels), severe over-segmentation occurs in 90% of SAM 2 cases but only 44% of SAM 3 cases. Multi-click prompting reduces these failure rates for both models, but SAM 2 still shows substantially higher severe-error frequencies (e.g., 73% vs. 65% in the smallest bin and 49% vs. 13% in the 2k–10k bin). Bounding-box prompts largely suppress over-segmentation in both models, with severe over-segmentation falling below $\sim$7% cases for SAM 2 and $\sim$4% cases for SAM 3 across all size bins. In this strong-prompt regime, the initialization advantage of SAM 3 becomes modest, confirming that SAM 2's instability is primarily a sparse-prompt phenomenon.

### 3.2. Full-Volume/Sequence Segmentation Accuracy

While prompt-frame accuracy captures initialization quality, clinical applications require accurate segmentation across full 3D volumes or complete temporal sequences. Full-volume DSC, therefore, reflects the combined effect of both initialization and propagation under SAM 2's memory-based architecture and SAM 3's redesigned tracking pathway. Table 2 summarizes structure-wise performance across all prompting regimes.

Across modalities, a consistent pattern emerges. Under sparse guidance (single- and multi-click prompting), SAM 3 generally achieves higher full-volume DSC than SAM 2 for most targets, indicating that its stronger prompt-frame initialization translates into better sequence-level performance, especially for structures such as vessels, gastrointestinal segments, and cardiac chambers. As prompt strength increases to bounding boxes and masks, this global advantage narrows: both models approach similar accuracy for many large, well-contrasted organs, and performance instead splits by anatomical type.

In this stronger-prompt regime, SAM 2 is frequently more competitive or superior for several organs like kidneys, spleen, bladder, reflecting more conservative propagation once a reliable initialization is provided. In contrast, SAM 3 retains an advantage for low-contrast, highly deformable, or tubular anatomy, where tracking stability is more challenging. Representative failure cases, such as MR bladder and SegThy thyroid/vascular targets, illustrate that excellent prompt-frame DSC can still collapse to near-zero full-volume DSC for one model while the other maintains stable masks. Given SegThy's click-prompt breakdown and counterintuitive propagation behavior for both models under stronger prompts, we provide a deeper dataset-specific analysis of the failure patterns and why full-volume DSC can remain non-trivial despite near-zero prompt-frame DSC in Appendix G. These discrepancies foreshadow the retention and over-propagation behaviour quantified by the failure-mode analysis.

### 3.3. Failure-Mode Analysis: Temporal Retention and Over-Propagation

Volume-level DSC aggregates initialization and propagation into a single number, but interactive workflows care about how masks evolve over time. Here we examine two temporal failure modes defined in Section 2.5: (i) *retention*, i.e., how quickly a well-initialized mask

Table 2: Full-volume/sequence DSC (%) for zero-shot segmentation across modalities and anatomical structures using single-click (1,0), multi-click (1,2), bounding-box, and mask prompts. For each pair, the higher DSC is shown in **bold**. Color shading in the table denotes statistical significance for the better model: $p < 0.001$, $0.001 < p < 0.05$, and no shading for $p > 0.05$.

| Modality | Structure | Click (1,0) | | Click (1,2) | | BBox | | Mask | |
|---|---|---|---|---|---|---|---|---|---|
| | | SAM 2 | SAM 3 | SAM 2 | SAM 3 | SAM 2 | SAM 3 | SAM 2 | SAM 3 |
| CT | Adrenal Gland (L) | 19.13 | **25.14** | 20.79 | **34.54** | **49.02** | 46.67 | **47.59** | 41.78 |
| | Adrenal Gland (R) | 8.93 | **11.28** | 10.18 | **19.86** | **45.52** | 44.86 | **44.41** | 39.53 |
| | Aorta | 68.71 | **78.72** | 72.07 | **81.17** | 68.41 | **74.58** | 67.22 | **73.42** |
| | Bladder | 3.63 | **10.32** | 6.56 | **10.93** | 10.60 | **12.03** | 9.72 | **11.61** |
| | Colon Tumor | 11.16 | **15.98** | 13.03 | **17.27** | 16.58 | **18.24** | 18.53 | **19.36** |
| | Duodenum | 25.68 | **30.68** | 26.92 | **32.36** | 31.34 | **33.23** | 32.78 | **34.35** |
| | Esophagus | 3.88 | **37.00** | 8.12 | **48.28** | 60.60 | **68.44** | 59.75 | **68.22** |
| | Gallbladder | 22.88 | **30.39** | 31.68 | **34.53** | **49.62** | 38.00 | **48.47** | 36.68 |
| | Inferior Vena Cava | 70.08 | **79.28** | 65.21 | **78.77** | 69.89 | **78.54** | 70.41 | **78.47** |
| | Kidney (L) | 58.72 | **59.35** | **66.18** | 61.38 | **75.75** | 64.70 | **76.67** | 64.43 |
| | Kidney (R) | 54.02 | **65.66** | 66.01 | **67.43** | **78.15** | 72.52 | **78.78** | 72.32 |
| | Liver | 44.18 | **65.85** | 52.10 | **71.53** | 67.72 | **74.94** | 67.21 | **74.99** |
| | Lung Tumor | 6.78 | **18.72** | 13.95 | **30.06** | **44.22** | 42.48 | **46.33** | 43.20 |
| | Pancreas | 19.24 | **32.71** | 23.93 | **34.87** | 28.00 | **33.93** | 27.38 | **33.73** |
| | Pancreas Tumor | 11.64 | **17.00** | 13.32 | **18.72** | 27.09 | **28.47** | 26.49 | **29.06** |
| | Portal & Splenic Veins | 27.70 | **31.44** | 31.00 | **31.63** | **36.55** | 34.05 | **34.69** | 34.33 |
| | Prostate | 2.56 | **7.24** | 5.32 | **7.58** | **11.88** | 8.70 | **9.26** | 8.75 |
| | Spleen | 46.20 | **57.13** | 56.51 | **59.77** | **74.25** | 63.03 | **74.96** | 62.56 |
| | Stomach | 36.80 | **50.34** | 45.73 | **52.07** | 49.43 | **53.84** | 49.42 | **55.86** |
| MR | Aorta | 42.58 | **58.67** | 46.05 | **63.01** | 40.83 | **58.59** | 42.12 | **58.30** |
| | Bladder | 0.48 | **7.49** | **55.49** | 6.80 | **76.91** | 7.29 | **47.90** | 6.34 |
| | Gallbladder | 17.92 | **19.03** | **26.15** | 25.06 | **44.60** | 30.19 | **43.74** | 30.57 |
| | Hippocampus (Ant) | 12.19 | **16.96** | 11.16 | **17.86** | 22.16 | **23.62** | 24.86 | **25.37** |
| | Hippocampus (Post) | 12.94 | **33.89** | 14.00 | **33.10** | 16.91 | **18.43** | **23.74** | 23.46 |
| | Kidney (L) | **45.19** | 44.44 | **51.67** | 48.77 | **58.07** | 49.23 | **56.81** | 48.07 |
| | Kidney (R) | 52.73 | **54.43** | 60.53 | 55.93 | **63.28** | 58.29 | **64.40** | 57.42 |
| | Left Atrium | 17.72 | **30.41** | 26.22 | **43.65** | 22.47 | **44.23** | 18.94 | **39.46** |
| | Left Ventricle | 80.38 | **93.17** | 74.32 | **92.54** | 88.21 | **93.62** | 89.88 | **94.21** |
| | Liver | 36.37 | **57.46** | 42.94 | **63.12** | 51.25 | **56.39** | 48.65 | **56.42** |
| | Myocardium | 36.92 | **78.24** | 39.56 | **74.10** | 52.95 | **72.88** | 82.46 | **84.75** |
| | Pancreas | 6.49 | **22.87** | 11.26 | **24.57** | 18.24 | **24.35** | 17.72 | **23.83** |
| | Prostate | 12.22 | **17.70** | 15.90 | **19.59** | **28.12** | 23.61 | **28.10** | 23.65 |
| | Right Ventricle | 52.39 | **77.29** | 46.01 | **82.76** | 80.41 | **85.60** | 82.61 | **86.37** |
| | Spleen | 26.91 | **49.94** | 36.22 | **56.99** | 56.52 | **58.78** | **59.15** | 59.11 |
| US | Carotid Artery (L) | **13.55** | 10.99 | **23.53** | 23.46 | 5.98 | **56.65** | 5.65 | **41.36** |
| | Carotid Artery (R) | 1.14 | **11.92** | 17.83 | **21.67** | 17.00 | **51.12** | 22.86 | **60.69** |
| | Jugular Vein (L) | 7.76 | **30.69** | 19.62 | **30.55** | 5.45 | **35.30** | 5.57 | **39.52** |
| | Jugular Vein (R) | 2.33 | **17.84** | 7.93 | **31.92** | 10.71 | **28.16** | 21.49 | **29.05** |
| | Left Atrium | 19.49 | **28.59** | 30.34 | **66.11** | 79.08 | **83.82** | 90.34 | **90.60** |
| | LV Endocardium | 27.47 | **67.48** | 62.50 | **72.98** | 85.38 | **85.79** | **91.93** | 91.19 |
| | LV Epicardium | 24.15 | **28.04** | 25.84 | **27.24** | 41.54 | **42.29** | **82.20** | 78.17 |
| | Thyroid | 10.13 | **32.65** | 19.87 | **53.90** | 10.53 | **28.10** | 7.27 | **27.58** |
| Endoscopy | Abdominal Wall | 55.77 | **67.42** | 58.14 | **79.41** | 69.20 | **82.56** | 81.23 | **87.81** |
| | Blood | 5.11 | **12.94** | 7.91 | **38.80** | 25.77 | **33.39** | 10.14 | **38.22** |
| | Connective Tissue | **70.45** | 66.32 | **65.73** | 61.16 | 61.14 | **72.24** | 69.91 | **74.55** |
| | Cystic Duct | **0.15** | 0.14 | **0.20** | 0.14 | **1.42** | 0.20 | **0.16** | 0.16 |
| | Fat | 60.00 | **71.43** | **61.71** | 60.90 | 38.42 | **40.78** | 87.20 | **88.14** |
| | Gallbladder | 72.49 | **79.18** | 74.95 | **75.38** | **82.81** | 81.17 | 78.73 | **83.80** |
| | GI Tract | 37.83 | **65.02** | 31.49 | **70.83** | 69.57 | **75.84** | **76.77** | 73.30 |
| | Grasper | 74.59 | **82.47** | 75.64 | **78.21** | 77.65 | **78.93** | 76.35 | **80.76** |
| | Hepatic Vein | 19.49 | **21.62** | 20.05 | **21.70** | 20.71 | **21.64** | 20.76 | **23.08** |
| | L-Hook Electrocautery | **66.84** | 64.50 | 65.91 | **68.58** | 65.91 | **69.64** | 66.78 | **69.98** |
| | Liver | 57.40 | **59.78** | 60.89 | **68.48** | **67.76** | 67.12 | 88.49 | **90.33** |
| | Liver Ligament | **98.69** | 98.29 | **98.65** | 96.16 | **98.76** | 98.55 | **98.72** | 98.51 |

drifts or degrades while the object is still present, and (ii) *over-propagation*, i.e., how long a model continues to hallucinate a mask after the object has disappeared.

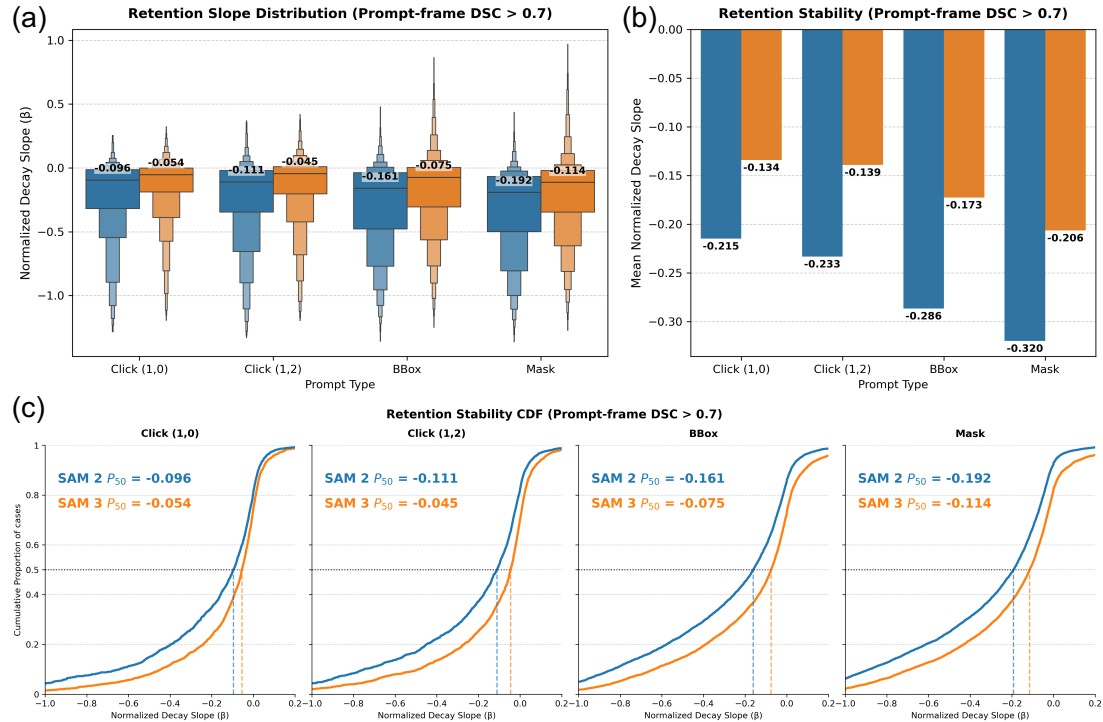

Figure 3: **Retention decay analysis for well-initialized cases.** Analysis is restricted to cases with $DSC \geq 0.7$ on the prompt frame. **(a)** Boxen plots showing the distribution of normalized decay slopes across prompting modes, where more negative values correspond to faster degradation in DSC as the object evolves over time. **(b)** Mean normalized decay slopes, summarizing the average retention behavior for each model and prompt type. **(c)** Cumulative distribution functions of the decay slopes, with annotated medians ($P_{50}$) highlighting that SAM 3 consistently exhibits less negative slopes than SAM 2. [Color: SAM 2, SAM 3]

**Retention of well-initialized objects.** To isolate propagation behaviour from pure initialization failures, we restrict this analysis to cases with good starting masks (prompt-frame $DSC \geq 0.7$), so that the decay slopes primarily reflect how well each model maintains a reasonable segmentation rather than how quickly an already-bad mask collapses. For completeness, we also report the same analysis computed over all cases (including poor initializations with prompt-frame $DSC < 0.7$) in Appendix H (Figure 8).

Figure 3 summarizes retention for cases with good initialization (prompt-frame $DSC \geq 0.7$). The boxen plots show the distribution of normalized decay slopes $\beta$ for each prompt type, where more negative values correspond to faster loss of accuracy from the first to the last frame. The bar plot reports mean slopes by prompt type, and the ECDF curves show, for any threshold on $\beta$, what fraction of cases have decay no worse than that value; the

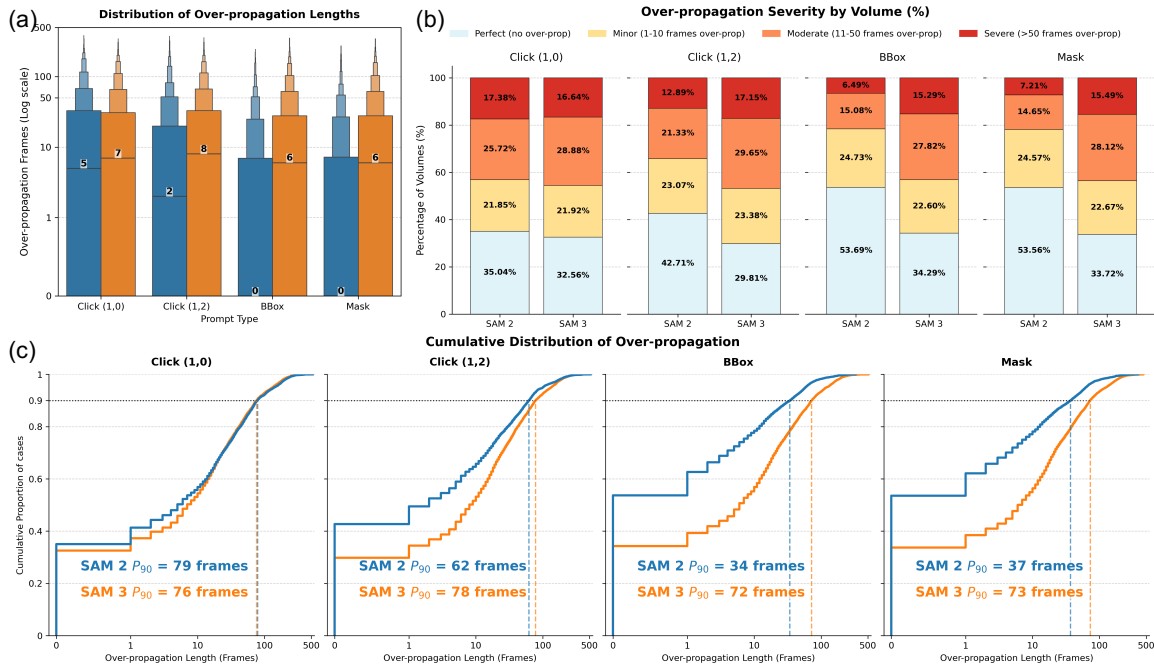

Figure 4: **Analysis of over-propagation after object disappearance.** Over-propagation length is defined as the number of frames with non-empty prediction after the last frame where the ground-truth object is present. **(a)** Boxen plots (log-scaled $y$-axis) showing the distribution of over-propagation lengths for each model and prompting mode. **(b)** Stacked bar plots summarizing the severity distribution across four bins: perfect termination, minor, moderate, and severe. **(c)** Cumulative distribution functions of over-propagation length, with annotated 90th-percentile values ($P_{90}$). [Color: SAM 2, SAM 3]

annotated $P_{50}$ markers indicate the median slope (half of the cases decay faster, half more slowly).

Across all well-initialized cases, both models exhibit negative normalized decay slopes on average, indicating that segmentation quality tends to deteriorate as the object evolves (Figure 3). However, SAM 3 consistently forgets more slowly. Under single-click prompts, the mean decay slope is $-0.215$ for SAM 2 versus $-0.134$ for SAM 3, and the median slopes ($P_{50}$) are $-0.096$ and $-0.054$, respectively, implying that a typical SAM 2 sequence loses roughly twice as much DSC over its lifespan as a typical SAM 3 sequence. Multi-click prompts show a similar pattern (mean slopes $-0.233$ vs. $-0.139$; medians $-0.111$ vs. $-0.045$). The difference widens with stronger prompts: for bounding boxes, mean slopes are $-0.286$ (SAM 2) and $-0.173$ (SAM 3), with medians of $-0.161$ and $-0.075$; for masks, the means are $-0.320$ vs. $-0.206$ and medians $-0.192$ vs. $-0.114$. In the ECDFs, SAM 3's curves are consistently shifted toward less negative values, indicating that, conditional on a good start, SAM 3 maintains segmentation quality better across all prompt types.

**Over-propagation after object disappearance.** Figure 4 reveals the cost of SAM 3's retention stability: a tendency to be "sticky." The distributions of over-propagation length highlight how many frames each model continues to predict foreground after the last ground-truth frame, the stacked bars group volumes into none/minor/moderate/severe hallucination (0, 1–10, 11–50, $> 50$ frames), and the ECDF curves describe the cumulative distribution of hallucinated length; the annotated $P_{90}$ gives the number of frames below which 90% of cases fall. Across prompt settings, the distribution is dominated by short or zero over-propagation for many cases, but a small fraction of failures persist for much longer durations, motivating tail-focused summaries in addition to $P_{90}$.

Under single-click (1,0) prompts, both models behave similarly: about 35% of SAM 2 volumes and 33% of SAM 3 volumes terminate perfectly with zero over-propagation, and the $P_{90}$ values are comparable (79 vs. 76 frames). This similarity also holds in central tendency (mean/median $L = 27.8/5$ for SAM 2 vs. 26.9/7 for SAM 3), while rare long failures remain (approximately $P_{99} = 243$ vs. 227 frames; $\sim 7.7\%$ vs. $\sim 7.1\%$ of cases exceed 100 over-propagated frames). The maximum observed failure is $L = 533$ frames, occurring in the CholecSeg8K dataset (Cystic Duct) under single-click prompting, and is observed for both models on the same case. With multi-click prompts, SAM 2 becomes slightly more conservative, with about 43% of volumes showing no over-propagation compared with about 30% for SAM 3, and $P_{90}$ dropping to 62 frames for SAM 2 versus 78 frames for SAM 3. Consistent with this shift, the bulk of the distribution tightens for SAM 2 (mean/median 20.5/2; $P_{99} \approx 215$; $\sim 5.0\% > 100$ frames), while SAM 3 retains a heavier tail (mean/median 27.7/8; $P_{99} \approx 228$; $\sim 7.4\% > 100$ frames).

The contrast is sharper once strong visual prompts is provided. For bounding-box prompts, 54% of SAM 2 volumes exhibit no over-propagation, compared with only 34% for SAM 3, and severe tails of more than 50 hallucinated frames occur in 6.5% of SAM 2 cases but 15.3% of SAM 3 cases; the corresponding $P_{90}$ values are 34 vs. 72 frames. Within the $> 50$ "severe" category, long failures remain more frequent for SAM 3 under bounding-box prompting: $\sim 1.74\%$ (SAM 2) vs. $\sim 6.56\%$ (SAM 3) exceed 100 over-propagated frames, with $P_{99} \approx 139$ vs. $\approx 212$ frames. Mask prompting shows a similar trend: roughly 54% of SAM 2 volumes versus 34% of SAM 3 volumes have zero over-propagation, while severe tails appear in 7.2% vs. 15.5% of cases and $P_{90}$ increases from 37 frames (SAM 2) to 73 frames (SAM 3). In this case, $\sim 2.11\%$ (SAM 2) vs. $\sim 6.56\%$ (SAM 3) exceed 100 frames, with $P_{99} \approx 158$ vs. $\approx 218$ frames.

Taken together with the prompt-frame over-segmentation analysis, these failure-mode results highlight a complementary trade-off between the models. SAM 3 offers more reliable initialization and better retention for well-initialized objects, particularly under stronger prompts, but is more "sticky" and prone to long-lived hallucinated masks after the object disappears. SAM 2 is less capable under sparse prompts and struggles with most targets, yet it tends to terminate tracks earlier and exhibits fewer extreme over-propagation failures under bounding-box and mask prompting.

### 3.4. Performance Behavior as a Function of Prompt Strength

Across modalities, both models follow a consistent pattern as prompt strength increases from single-click to multi-click, bounding-box, and mask prompts. Under sparse guidance (clicks),

SAM 3 dominates because it interprets minimal prompts more reliably, leading to higher prompt-frame DSC, fewer prompt-frame over-segmentation failures, and better temporal retention. As prompts become more informative and provide explicit spatial support, the global advantage narrows and performance becomes modality and structure-dependent: SAM 2 is often competitive or better on targets such as kidneys, spleen, and bladder under bounding-box/mask prompting in CT/MRI, whereas SAM 3 more consistently leads on several vessel and tract targets (e.g., aorta/IVC/portal-venous structures and GI tract) and shows strong gains on challenging ultrasound targets (e.g., SegThy thyroid). Qualitative examples illustrating these regimes are shown in Figures 5 and 6.

### 3.5. Overall Interpretation and Summary of Findings

Taken together, the prompt-frame, full-volume, and failure-mode evaluations show that SAM 2 and SAM 3 offer complementary strengths rather than a single performance hierarchy, driven by a trade-off between prompt interpretation (what to segment) and temporal consistency (how well it is remembered). We summarize our findings as follows:

- **Initialization advantages for SAM 3.** Under click prompts, SAM 3 has a clear advantage: its unified perception encoder infers structure from minimal input, yielding higher prompt-frame DSC and substantially fewer flooding failures than SAM 2 across most targets.

- **Propagation trade-offs under strong visual prompts.** Under bounding-box or mask prompts, SAM 2 is often competitive or better on kidneys, spleen, and bladder in CT/MRI, and it more frequently terminates tracks without long over-propagation failures. SAM 3 remains stronger on several vessel/tract targets (e.g., aorta/IVC/portal-venous structures and GI tract) and on challenging ultrasound targets such as SegThy thyroid, but it more often exhibits longer over-propagation tails.

- **The "unreliable propagator" risk.** High initialization accuracy does not guarantee successful propagation. In several datasets (e.g., MR bladder, SegThy ultrasound), one model attains excellent prompt-frame DSC but then collapses or hallucinates for many frames. This highlights the need to evaluate temporal retention and over-propagation beyond prompt-frame or volume-averaged DSC.

Overall, SAM 3 is the natural default for interactive zero-shot segmentation on 3D medical data. Across modalities, it is markedly more reliable under sparse click prompts due to stronger prompt interpretation and more stable retention, and these advantages frequently translate into better full-volume performance. As the visual prompt becomes stronger (bounding box or mask), the performance gap narrows for many targets; however, SAM 3 remains competitive in this regime as well, and continues to lead on a broad set of structures. The main exceptions are a smaller subset of targets under bounding-box or mask initialization where SAM 2 achieves higher Dice and behaves more conservatively with fewer long over-propagation tails.

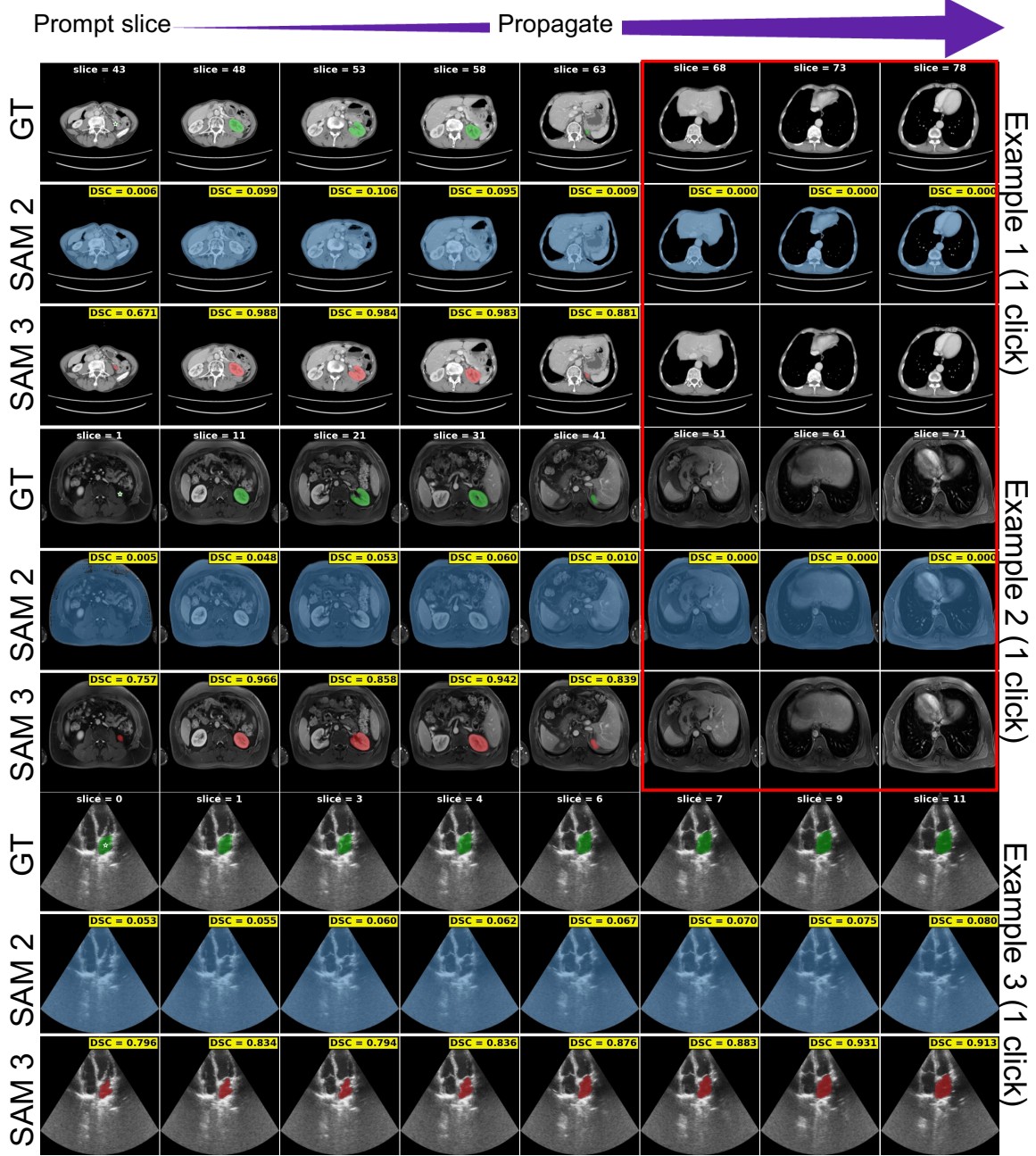

Figure 5: **Qualitative examples where SAM 3 outperforms SAM 2**. All examples show SAM 3's superior prompt initialization by better localizing the target even under sparse prompts, whereas SAM 2 fails to localize the target on the prompted frame and across the volume/sequence, resulting in flooding/over-segmentation and notably lower DSC. Moreover, in examples 1–2, the red-boxed columns (Example 1: slice $\geq$ 68; Example 2: slice $\geq$ 51) correspond to slices where the GT mask is empty (target absent); SAM 2 continues to produce residual masks beyond the last object frame, while SAM 3 terminates more cleanly. [Colors: GT, SAM 2, SAM 3]

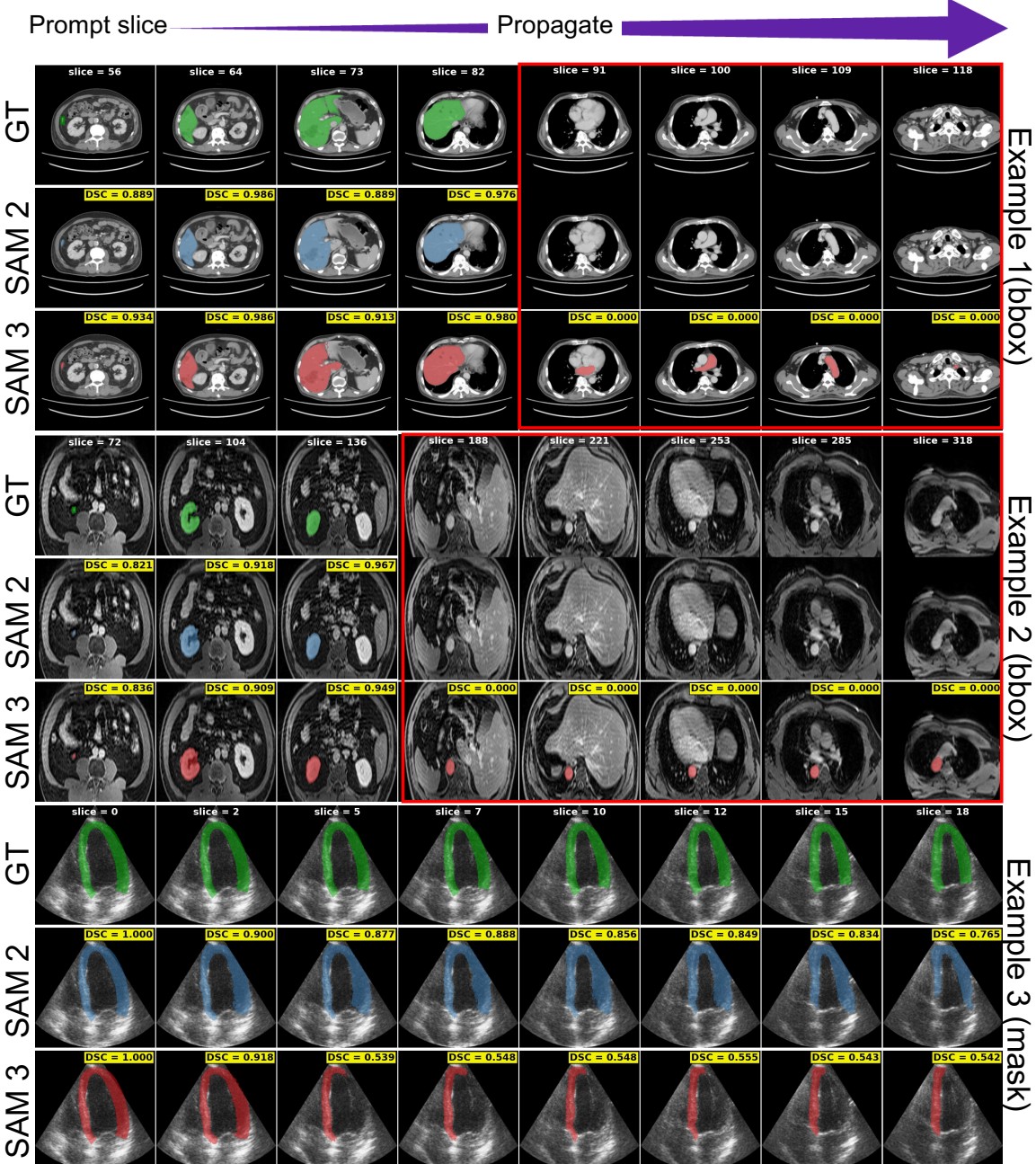

Figure 6: **Qualitative examples where SAM 2 outperforms SAM 3**. In all three examples, SAM 3 shows stronger initial localization but exhibits propagation failures, including hallucinated residual masks in later slices (Examples 1–2) and erosion/collapse of structure boundaries under challenging appearance or motion (Example 3). In the red-boxed columns of Examples 1–2 (Example 1: slice ≥ 91; Example 2: slice ≥ 188), the GT mask is empty (target absent); SAM 2 correctly predicts no mask, whereas SAM 3 produces residual masks, indicating over-propagation. [Colors: GT, SAM 2, SAM 3]

## 4. Conclusion

This work presents the first large-scale, controlled comparison of SAM 2 and SAM 3 for zero-shot segmentation of 3D medical data under identical visual prompting. By evaluating single-click, multi-click, bounding-box, and mask initialization across sixteen datasets and 54 anatomical structures, we disentangle how architectural changes in SAM 3 affect prompt interpretation, temporal retention, and failure behaviour relative to SAM 2.

Our results show that SAM 3 offers markedly stronger prompt interpretation: it delivers higher prompt-frame DSC, substantially fewer over-segmentation failures, and slower temporal decay of prediction mask for well-initialized objects, especially under click and bounding-box prompts. SAM 2, however, remains a competitive and often preferable choice for some organs like kidney, gallbladder, spleen in CT/MRI under strong visual prompts, where its propagation is more conservative and less prone to long-lived hallucinated masks. The failure-mode analysis highlights that high initialization accuracy alone is not sufficient: models can still suffer catastrophic collapse or prolonged over-propagation, underscoring the need to explicitly evaluate temporal retention and termination behaviour.

Overall, our findings position SAM 3 as the stronger default backbone for broad 3D medical segmentation workflows, while clarifying scenarios in which SAM 2 remains the safer propagator for specific organ types and prompt regimes. A key limitation of this study is that we restrict the comparison to purely visual prompts, deliberately disabling the concept- and text-based mechanisms introduced in SAM 3. As vision–language approaches such as Voxtell (Rokuss et al., 2025) gain traction for open-vocabulary 3D medical segmentation, extending our framework to include semantic prompting and language-guided concepts, with SAM 3's full capabilities enabled, is an important direction for future work.

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

## Appendix A. Scope of comparison vis-à-vis supervised baselines

The main study is a controlled, zero-shot comparison of SAM 2 and SAM 3 models on 3D medical data under identical visual prompts. This design isolates differences in prompt interpretation and propagation dynamics and avoids conflating model behavior with medical training data, dataset curation, or task-specific optimization.

Under this scope, models trained on labeled medical data, whether framed as 3D medical foundation models (e.g., MedSAM2 (Ma et al., 2025)) or task-specific supervised models (e.g., nnU-Net (Isensee et al., 2021)), fall outside the study setting because they are not zero-shot and their performance reflects domain training and dataset choices in addition to architecture. We also exclude 2D-only promptable models such as MedSAM (Ma et al., 2024a), since they do not natively support 3D data. Incorporating them into our evaluation would require adding an external propagation mechanism, introducing an additional algorithmic component that would confound attribution. Moreover, task-specific models like nnU-Net (Isensee et al., 2021) are typically trained per dataset, whereas our evaluation targets a single promptable model that can be applied uniformly across datasets and targets without retraining.

Beyond the general protocol mismatch, MedSAM2 introduces two practical comparability issues. First, it is commonly restricted to a bounding-box-centric prompting interface, which does not align with our multi-prompt evaluation and the corresponding analyses of prompt strength. Second, rigorous multi-dataset benchmarking benefits from clear documentation of training dataset composition and splits to verify overlap assumptions; at the time of our study, these details were not fully accessible for MedSAM2 in a way that supports strict overlap verification for our evaluation. This limited transparency complicates data provenance checks, i.e., tracing exactly which source datasets and subsets contributed to training, making it difficult to rule out inadvertent overlap or leakage when evaluating across multiple public benchmarks.

## Appendix B. SAM 3 PVS configuration details and implications

This appendix documents the SAM 3 configuration used in our experiments and clarifies how it follows the usage described in the SAM 3 paper. SAM 3 distinguishes Promptable Visual Segmentation (PVS), where targets are specified by visual prompts (points, boxes, masks), from concept-driven prompting mechanisms that use semantic inputs (Carion et al., 2025). Our evaluation uses the PVS setting so that SAM 2 and SAM 3 are compared under the same interaction pattern: a first-frame visual prompt followed by forward propagation through the sequence.

We implement SAM 3 in PVS by using the authors' released tracker-based visual-prompt interface, i.e., by selecting the corresponding inference path rather than modifying parameters or introducing custom bypasses. Concretely, for all SAM 3 inferences done in this work, we build upon the official SAM 3 implementation that exposes a SAM 2-style video task interface[1]. For all SAM 2 inferences, we use the official SAM 2 VOS script as the

---

1. https://github.com/facebookresearch/sam3/blob/11dec2936de97f2857c1f76b66d982d5a001155d/examples/sam3_for_sam2_video_task_example.ipynb

reference [2]. Under this implementation of SAM 3, we provide only visual prompts on the initialization frame and run the released forward propagation loop; concept prompts (text or exemplar inputs) are not provided, and concept/PCS handlers are not invoked. As a result, predictions are determined by SAM 3's Perception Encoder features, the prompt-conditioned initialization on the prompt frame, and the tracker's temporal propagation and memory updates over subsequent frames.

This choice keeps the comparison controlled and aligned with the intended PVS usage: both SAM 2 and SAM 3 operate from the same class of visual inputs and the same interaction protocol, and the evaluation isolates differences in visual prompt interpretation and propagation dynamics without introducing additional implementation degrees of freedom. Over-propagation is measured from the masks produced by this standard PVS propagation after the final ground-truth frame, and therefore reflects termination behavior of the tracker under visual-only interaction, rather than behavior induced by any custom parameter masking or auxiliary semantic inputs.

## Appendix C. SAM 2 vs. SAM 3 under visual prompting

This appendix summarizes the SAM 2 vs. SAM 3 design differences that are most relevant under the PVS regime studied in this paper, where the same visual-prompt protocol is applied and performance is assessed through long-horizon propagation over 3D medical volumes/cines. The behaviors quantified in the main text are governed by two coupled factors: the learned per-frame visual representation available to the tracking stack (which shapes how sparse prompt evidence is resolved on the initialization frame) and the temporal conditioning dynamics induced by the propagation/memory formulation (which shapes retention, drift, and termination after disappearance).

In SAM 2, per-frame representations are produced by a Hiera backbone and propagation is implemented as memory-conditioned decoding over a streaming memory bank: a memory encoder writes features/masks into memory and a memory-attention module conditions current-frame decoding on this bank during propagation (Ravi et al., 2024; Ryali et al., 2023).

In SAM 3, the tracker consumes Perception Encoder (PE) features that are shared with the detector/tracker stack in the official design, and the overall system adopts a detector–tracker factorization in which a DETR-style detector with learned object queries provides object-centric localization/association while the tracker inherits the SAM 2 transformer encoder–decoder for video segmentation and interactive refinement (Carion et al., 2025, 2020). Importantly, for the PVS setting, the tracker in SAM 3 retains a SAM 2-style propagation mechanism with a memory encoder and memory bank (Carion et al., 2025); however, even when both models employ memory banks, the effective temporal update dynamics can differ because the tracker operates on different learned features (Hiera vs. PE) and because the detector–tracker decomposition introduces stronger object-centric identity and association priors that couple to how state is written, retrieved, and maintained over time. Consequently, the balance between prompt evidence and model priors on the initial-

---

2. https://github.com/facebookresearch/sam2/blob/2b90b9f5ceec907a1c18123530e92e794ad901a4/tools/vos_inference.py

ization frame, as well as long-horizon stability and termination behavior, can differ across SAM 2 and SAM 3 under identical visual prompting.

These architectural differences relate to the three error modes quantified in the main text. Prompt-frame over-segmentation is most sensitive to the per-frame feature space and the initialization pathway, since sparse prompts must be resolved into a precise object extent on the prompt frame. Temporal retention is most sensitive to temporal conditioning and memory updates while the target remains present, since small initialization errors may be reinforced or corrected depending on what is written to memory and how strongly it is reused during propagation. Over-propagation after object disappearance is most sensitive to termination behavior and persistence of tracker/memory state after visual evidence vanishes, where differences in identity persistence induced by the tracker formulation can translate into different stopping characteristics. Since the visual prompting protocol is held constant across models in our study, differences in model behavior are most naturally interpreted as consequences of differences in learned visual representations, initialization pathway, and long-horizon propagation/termination dynamics between SAM 2 and SAM 3 under the same PVS interaction protocol.

## Appendix D. Click prompting and robustness to annotation variability

### D.1. Click prompting protocol

To enable a controlled SAM 2 vs. SAM 3 comparison under click prompting, we generate prompts in a model-independent manner and reuse identical prompt coordinates for both models. We therefore avoid oracle-style interactive protocols that place later clicks based on model-specific error regions, which would yield different click sequences across models. Instead, we define all clicks solely from the ground-truth mask on the initialization frame and then propagate without further interaction.

For each target object, we define an initialization frame $t_0$ as the first frame in which the ground-truth mask is non-empty. Given the binary ground-truth mask $M_{gt}^{(t_0)}$ on the initialization frame, we place the initial positive click at the most interior point of the object following the SAM 2 and SAM 3 prompt protocol, defined as the pixel attaining the maximum of the Euclidean distance transform of $M_{gt}^{(t_0)}$ (i.e., the foreground pixel farthest from the mask boundary). This yields a deterministic positive click location per object.

For the (1,2) setting, we add two negative clicks sampled from a local neighborhood around the object. We define a neighborhood ring as the set difference between a dilated mask and the original foreground mask, i.e., $\mathcal{N} = \mathrm{dilate}(M_{gt}^{(t_0)}) \setminus M_{gt}^{(t_0)}$ with a fixed dilation radius. The first negative click is selected as the point in $\mathcal{N}$ farthest from the object centroid, and the second negative click is selected via a maximin criterion to be as far as possible from the first negative click (while remaining in $\mathcal{N}$). This procedure is deterministic given $M_{gt}^{(t_0)}$ and produces the same negative click locations for both models.

All click coordinates are saved per case and reused across SAM 2 and SAM 3. Consequently, any performance differences under click prompting reflect differences in prompt interpretation and propagation rather than differences in prompt placement.

Table 3: Robustness to click jitter under single-click (1,0) and multi-click (1,2) prompting. We use jitter radius $J = \pm 5$ pixels and $K = 5$ jitter trials per case. For each dataset/structure, we report expected DSC under jitter as $\mu_{\text{jitter}} \pm \sigma_{\text{jitter}}$ (baseline), where $\mu_{\text{jitter}}$ is the mean DSC under jitter averaged over cases, $\sigma_{\text{jitter}}$ is the *average within-case* standard deviation across jitter trials (jitter sensitivity), and baseline is the canonical-click DSC mean across cases. All values are in %.

| Dataset | Structure | Click (1,0) | | Click (1,2) | |
|---|---|---|---|---|---|
| | | SAM 2 | SAM 3 | SAM 2 | SAM 3 |
| BTCV | Adrenal Gland (L) | $6.8_{\pm1.5}$ (6.9) | $16.3_{\pm4.1}$ (15.6) | $9.2_{\pm1.7}$ (9.7) | $26.3_{\pm7.1}$ (22.7) |
| | Adrenal Gland (R) | $2.6_{\pm2.0}$ (3.7) | $13.3_{\pm6.2}$ (14.5) | $5.3_{\pm2.2}$ (6.9) | $17.1_{\pm5.6}$ (17.3) |
| | Aorta | $88.0_{\pm0.1}$ (88.0) | $90.6_{\pm0.2}$ (90.6) | $87.1_{\pm0.6}$ (87.2) | $90.8_{\pm0.2}$ (90.8) |
| | Esophagus | $2.7_{\pm1.5}$ (5.1) | $50.4_{\pm5.3}$ (52.9) | $9.6_{\pm1.9}$ (9.9) | $54.4_{\pm8.3}$ (56.0) |
| | Gallbladder | $22.6_{\pm4.5}$ (19.4) | $30.9_{\pm2.7}$ (32.1) | $32.3_{\pm6.2}$ (31.9) | $33.3_{\pm4.1}$ (31.1) |
| | Inferior Vena Cava | $63.4_{\pm1.3}$ (64.6) | $75.6_{\pm1.2}$ (74.9) | $58.6_{\pm2.7}$ (58.6) | $75.5_{\pm0.3}$ (75.5) |
| | Kidney (L) | $55.0_{\pm4.4}$ (60.4) | $54.1_{\pm5.8}$ (54.7) | $70.1_{\pm4.8}$ (71.6) | $64.6_{\pm4.4}$ (67.5) |
| | Kidney (R) | $57.8_{\pm3.3}$ (56.4) | $67.6_{\pm4.0}$ (66.0) | $66.8_{\pm4.0}$ (68.2) | $68.6_{\pm5.2}$ (70.4) |
| | Liver | $45.0_{\pm4.3}$ (45.2) | $68.8_{\pm4.9}$ (71.5) | $52.6_{\pm3.8}$ (52.2) | $77.6_{\pm1.9}$ (80.0) |
| | Pancreas | $20.4_{\pm4.3}$ (22.9) | $34.8_{\pm3.8}$ (35.0) | $25.2_{\pm5.1}$ (27.4) | $38.4_{\pm4.1}$ (40.8) |
| | Portal & Splenic Veins | $29.2_{\pm2.9}$ (27.7) | $31.4_{\pm3.1}$ (31.4) | $31.9_{\pm3.8}$ (31.0) | $33.0_{\pm3.8}$ (31.6) |
| | Spleen | $56.8_{\pm2.5}$ (57.5) | $61.1_{\pm1.5}$ (59.2) | $66.2_{\pm2.0}$ (65.8) | $61.4_{\pm2.5}$ (59.6) |
| | Stomach | $33.5_{\pm6.3}$ (27.0) | $48.9_{\pm5.0}$ (45.7) | $38.6_{\pm3.3}$ (39.2) | $49.6_{\pm4.5}$ (49.4) |
| FLARE22 | Adrenal Gland (L) | $24.5_{\pm3.9}$ (26.4) | $27.5_{\pm4.1}$ (30.9) | $27.3_{\pm3.1}$ (27.4) | $42.3_{\pm6.4}$ (41.6) |
| | Adrenal Gland (R) | $8.9_{\pm4.3}$ (12.1) | $10.8_{\pm4.3}$ (9.3) | $12.7_{\pm4.8}$ (12.2) | $21.5_{\pm10.0}$ (21.4) |
| | Aorta | $93.3_{\pm0.1}$ (93.3) | $95.9_{\pm0.1}$ (95.9) | $93.5_{\pm0.9}$ (93.8) | $95.9_{\pm0.1}$ (95.9) |
| | Duodenum | $25.7_{\pm2.1}$ (25.7) | $32.4_{\pm3.8}$ (30.7) | $28.2_{\pm4.3}$ (26.9) | $33.9_{\pm2.5}$ (32.4) |
| | Esophagus | $4.2_{\pm1.1}$ (3.2) | $31.6_{\pm4.1}$ (27.5) | $6.4_{\pm1.5}$ (7.1) | $41.7_{\pm7.2}$ (43.6) |
| | Gallbladder | $35.9_{\pm2.1}$ (36.0) | $47.2_{\pm2.9}$ (47.6) | $38.4_{\pm5.5}$ (39.9) | $48.8_{\pm4.5}$ (50.0) |
| | Inferior Vena Cava | $73.4_{\pm0.3}$ (73.4) | $81.7_{\pm0.5}$ (81.9) | $70.1_{\pm2.6}$ (69.2) | $82.0_{\pm0.3}$ (80.7) |
| | Kidney (L) | $80.2_{\pm2.8}$ (82.0) | $78.4_{\pm1.0}$ (78.1) | $82.0_{\pm2.8}$ (78.9) | $78.8_{\pm1.9}$ (78.6) |
| | Kidney (R) | $85.5_{\pm0.9}$ (86.0) | $87.2_{\pm1.4}$ (87.7) | $86.5_{\pm1.3}$ (86.6) | $89.9_{\pm2.1}$ (90.5) |
| | Liver | $72.2_{\pm3.5}$ (73.6) | $86.6_{\pm1.6}$ (86.3) | $79.6_{\pm0.6}$ (80.0) | $87.5_{\pm1.6}$ (86.5) |
| | Pancreas | $28.4_{\pm2.7}$ (28.2) | $40.5_{\pm4.8}$ (38.8) | $32.2_{\pm4.8}$ (32.9) | $45.2_{\pm6.1}$ (44.4) |
| | Spleen | $71.3_{\pm2.1}$ (70.9) | $72.9_{\pm1.1}$ (72.4) | $78.5_{\pm3.5}$ (79.1) | $75.3_{\pm0.8}$ (75.4) |
| | Stomach | $44.2_{\pm4.0}$ (42.7) | $51.9_{\pm2.6}$ (53.1) | $48.8_{\pm6.1}$ (49.7) | $55.3_{\pm2.5}$ (53.6) |
| MSD Heart | Left Atrium | $17.6_{\pm0.8}$ (17.7) | $35.5_{\pm8.6}$ (30.4) | $25.2_{\pm3.4}$ (26.2) | $43.4_{\pm9.6}$ (43.7) |
| CAMUS | Left Atrium | $19.5_{\pm0.0}$ (19.5) | $28.1_{\pm0.9}$ (28.6) | $30.2_{\pm1.2}$ (30.3) | $65.9_{\pm2.8}$ (66.1) |
| | LV Endocardium | $27.4_{\pm0.0}$ (27.5) | $67.8_{\pm1.3}$ (67.5) | $62.5_{\pm2.0}$ (62.5) | $72.5_{\pm2.2}$ (73.0) |
| | LV Epicardium | $24.2_{\pm0.1}$ (24.2) | $28.2_{\pm1.7}$ (28.0) | $26.2_{\pm0.8}$ (25.8) | $26.8_{\pm2.2}$ (27.2) |
| CholecSeg8K | Abdominal Wall | $55.5_{\pm0.1}$ (55.8) | $69.5_{\pm2.8}$ (67.4) | $57.3_{\pm2.4}$ (58.1) | $78.3_{\pm2.5}$ (79.4) |
| | Blood | $11.0_{\pm0.2}$ (5.1) | $13.5_{\pm0.4}$ (12.9) | $8.0_{\pm0.1}$ (7.9) | $40.9_{\pm2.9}$ (38.8) |
| | Connective Tissue | $70.4_{\pm0.1}$ (70.5) | $66.3_{\pm0.1}$ (66.3) | $65.8_{\pm0.1}$ (65.7) | $61.2_{\pm0.2}$ (61.2) |
| | Cystic Duct | $0.1_{\pm0.0}$ (0.1) | $0.2_{\pm0.1}$ (0.1) | $0.2_{\pm0.0}$ (0.2) | $0.1_{\pm0.0}$ (0.1) |
| | Fat | $60.7_{\pm0.8}$ (60.0) | $71.5_{\pm0.4}$ (71.4) | $60.7_{\pm2.2}$ (61.7) | $61.0_{\pm3.9}$ (60.9) |
| | Gallbladder | $71.7_{\pm2.2}$ (72.5) | $74.9_{\pm2.8}$ (79.2) | $75.3_{\pm1.3}$ (75.0) | $79.5_{\pm3.1}$ (75.4) |
| | GI Tract | $38.7_{\pm1.3}$ (37.8) | $66.1_{\pm4.3}$ (65.0) | $32.5_{\pm1.8}$ (31.5) | $70.9_{\pm2.0}$ (70.8) |
| | Grasper | $74.8_{\pm0.2}$ (74.6) | $80.5_{\pm1.9}$ (82.5) | $77.7_{\pm2.3}$ (75.6) | $79.1_{\pm2.0}$ (78.2) |
| | Hepatic Vein | $19.5_{\pm0.0}$ (19.5) | $21.6_{\pm0.1}$ (21.6) | $19.8_{\pm0.1}$ (20.1) | $21.8_{\pm0.0}$ (21.7) |
| | L-Hook Electrocautery | $66.8_{\pm0.1}$ (66.8) | $68.3_{\pm2.3}$ (64.5) | $65.6_{\pm0.3}$ (65.9) | $68.5_{\pm0.8}$ (68.6) |
| | Liver | $57.9_{\pm2.1}$ (57.4) | $63.7_{\pm5.6}$ (59.8) | $59.3_{\pm1.4}$ (60.9) | $66.3_{\pm3.1}$ (68.5) |
| | Liver Ligament | $98.7_{\pm0.0}$ (98.7) | $98.3_{\pm0.0}$ (98.3) | $98.6_{\pm0.0}$ (98.6) | $97.4_{\pm0.8}$ (96.2) |

**D.2. Robustness to annotation noise via click jitter**

To assess the sensitivity of click prompting to realistic annotation variability while maintaining identical prompts across models, we perform a click-jitter experiment in which the initialization prompt set is recomputed under small random perturbations. For each case, we begin from the original non-jitter positive click ("canonical click") on the initialization frame $t_0$ and sample independent spatial offsets $\delta_x, \delta_y \sim \text{Unif}(-J, J)$ to obtain a jittered positive click. For multi-click prompting (1,2), negative clicks are recomputed conditioned on the jittered positive click using the same ring-based protocol described in Appendix D.1, ensuring that the entire prompt set varies coherently with the user's initial placement. We re-run inference under $K$ jitters per case.

For each case, we summarize jitter robustness by the mean DSC across jitters ($\mu_{\text{jitter}}$), and the within-case standard deviation across jitters ($\sigma_{\text{jitter}}$). We then aggregate these values on a dataset/structure level by reporting both $\mu_{\text{jitter}}$ and $\sigma_{\text{jitter}}$ averaged over cases (jitter sensitivity). For reference, we also report the original canonical click DSC as a baseline. We report this sensitivity analysis on a subset of datasets spanning all four modalities: BTCV and FLARE22 (CT), MSD Heart (MR), CAMUS (ultrasound cine), and CholecSeg8K (endoscopy). All results are presented in Table 3.

## Appendix E. Preprocessing Details

All datasets were converted into a unified slice-based format to enable consistent evaluation across models and modalities. For CT datasets, images were first windowed using clinically standard ranges (e.g., soft-tissue windowing with level–width of 40/400 for abdominal CT and lung windowing of $-600/1500$ for thoracic CT) before clipping and rescaling to $[0, 255]$. MRI volumes were normalized by extracting the intensity values between the 0.5th and 99.5th percentiles within each volume and linearly rescaling this clipped range to $[0, 255]$, ensuring robustness to modality-specific dynamic range differences. Ultrasound images were min–max normalized per sequence and similarly rescaled to $[0, 255]$. Endoscopy datasets provided color-coded semantic masks, which were converted into per-class binary masks via RGB-to-class lookup. No smoothing, interpolation, or artifact removal was applied. All inputs are provided at native resolution and internally resized by the official code before encoding ($1024 \times 1024$); we follow the authors' released inference pipelines to ensure a faithful and reproducible comparison. This standardized preprocessing ensures consistent inputs across modalities, with any model-specific resizing handled internally by the official pipelines.

All experiments use publicly released checkpoints without any fine-tuning. For SAM 2, we use the SAM 2.1 Hiera-B+ checkpoint. The SAM 3 release does not specify multiple model variants in the paper, and we therefore adopt the standard configuration provided by the authors. All evaluations were performed on NVIDIA H100 GPUs.

## Appendix F. Detailed Prompt-Frame Results

This appendix provides the structure-wise breakdown of prompt-frame accuracy to supplement the analysis in Section 3.1. Table 4 summarizes the DSC across all structures, modalities, and prompt types. In CT, the gains for SAM 3 are substantial for anatom-

Table 4: Prompt–frame DSC (%) for zero-shot segmentation across modalities and anatomical structures using single-click (1,0), multi-click (1,2), and bounding-box prompts. For each pair, the higher DSC is shown in **bold**. Color shading in the table denotes statistical significance for the better model: $p < 0.001$, $0.001 < p < 0.05$, and no shading for $p > 0.05$.

| Modality | Structure | Click (1,0) | | Click (1,2) | | BBox | |
|---|---|---|---|---|---|---|---|
| | | SAM 2 | SAM 3 | SAM 2 | SAM 3 | SAM 2 | SAM 3 |
| CT | Adrenal Gland (L) | 20.28 | **25.56** | 25.81 | **37.11** | 77.04 | **78.38** |
| | Adrenal Gland (R) | 9.60 | **10.65** | 15.56 | **20.14** | 77.45 | **79.91** |
| | Aorta | 60.30 | **68.14** | 65.54 | **69.59** | 84.05 | **86.20** |
| | Bladder | 10.33 | **50.53** | 22.13 | **59.97** | 84.49 | **87.51** |
| | Colon Tumor | 25.45 | **44.05** | 35.21 | **52.08** | 74.76 | **76.82** |
| | Duodenum | 47.65 | **52.93** | 52.80 | **61.04** | 82.07 | **85.77** |
| | Esophagus | 5.20 | **33.93** | 20.68 | **46.13** | 85.97 | **86.85** |
| | Gallbladder | 24.38 | **44.24** | 37.32 | **52.06** | 82.68 | **84.38** |
| | Inferior Vena Cava | 86.25 | **90.79** | 84.81 | **91.24** | 91.25 | **93.63** |
| | Kidney (L) | 51.79 | **65.32** | 60.48 | **67.76** | 84.48 | **87.78** |
| | Kidney (R) | 49.94 | **65.13** | 61.59 | **67.24** | 84.67 | **88.17** |
| | Liver | 30.62 | **52.88** | 44.08 | **57.99** | 78.42 | **81.99** |
| | Lung Tumor | 5.08 | **19.21** | 20.12 | **29.90** | 75.89 | **76.62** |
| | Pancreas | 17.96 | **36.65** | 28.60 | **41.79** | 78.72 | **81.22** |
| | Pancreas Tumor | 22.01 | **37.43** | 28.10 | **42.81** | 85.55 | **88.24** |
| | Portal & Splenic Veins | 60.85 | **76.99** | 66.13 | **76.82** | 86.91 | **88.50** |
| | Prostate | 9.06 | **36.79** | 25.03 | **47.22** | 86.01 | **87.25** |
| | Spleen | 37.91 | **60.77** | 54.95 | **65.60** | 80.81 | **85.09** |
| | Stomach | 45.89 | **62.82** | 56.87 | **69.60** | 84.40 | **87.21** |
| MR | Aorta | 29.88 | **38.59** | 37.97 | **40.79** | 83.66 | **84.97** |
| | Bladder | 13.42 | **86.99** | 84.28 | **87.97** | **92.42** | 90.53 |
| | Gallbladder | 16.25 | **26.63** | 28.62 | **33.63** | 82.00 | **83.32** |
| | Hippocampus (Ant) | 6.89 | **18.70** | **23.69** | 20.90 | 82.12 | **82.31** |
| | Hippocampus (Post) | 3.16 | **8.92** | 6.14 | **13.37** | **82.82** | 79.41 |
| | Kidney (L) | 41.18 | **44.71** | 48.50 | **49.75** | 81.88 | **82.86** |
| | Kidney (R) | 40.84 | **49.34** | 49.83 | **52.22** | 82.26 | **84.88** |
| | Left Atrium | 4.58 | **9.97** | 15.16 | **18.30** | 75.25 | **79.67** |
| | Left Ventricle | 88.08 | **95.64** | 89.10 | **94.34** | 96.06 | **96.75** |
| | Liver | 19.82 | **34.04** | 29.60 | **39.75** | 76.87 | **78.63** |
| | Myocardium | 36.61 | **79.95** | 44.74 | **72.40** | 53.80 | **74.25** |
| | Pancreas | 4.20 | **17.38** | 16.13 | **23.58** | 75.27 | **78.24** |
| | Prostate | 13.82 | **29.82** | 26.80 | **35.58** | 84.08 | **84.39** |
| | Right Ventricle | 73.64 | **86.55** | 74.90 | **89.23** | 95.07 | **95.61** |
| | Spleen | 20.78 | **39.74** | 38.06 | **49.25** | 77.78 | **79.50** |
| US | Carotid Artery (L) | 0.38 | **0.68** | **3.62** | 1.61 | **64.05** | 57.98 |
| | Carotid Artery (R) | 0.18 | **1.46** | **14.53** | 3.30 | **61.06** | 60.97 |
| | Jugular Vein (L) | 1.14 | **2.65** | **6.74** | 2.93 | **58.41** | 55.61 |
| | Jugular Vein (R) | 0.40 | **1.63** | **6.99** | 2.50 | **58.32** | 54.97 |
| | Left Atrium | 17.61 | **25.31** | 47.37 | **60.30** | 76.99 | **82.61** |
| | LV Endocardium | 31.73 | **70.46** | 69.49 | **73.01** | **86.77** | 86.16 |
| | LV Epicardium | 23.98 | **27.65** | **30.73** | 26.04 | 34.15 | **34.93** |
| | Thyroid | 0.99 | **3.89** | **5.88** | 3.70 | **67.19** | 65.14 |
| Endoscopy | Abdominal Wall | 58.07 | **72.06** | 77.34 | **80.86** | **87.33** | 87.24 |
| | Blood | 3.06 | **3.08** | 42.03 | **63.01** | **73.81** | 72.91 |
| | Connective Tissue | **66.34** | 54.60 | **68.74** | 67.61 | 85.01 | **88.95** |
| | Cystic Duct | 30.50 | **32.92** | **29.70** | 13.45 | 41.92 | **48.84** |
| | Fat | 62.33 | **74.23** | **72.84** | 62.25 | **53.74** | 52.80 |
| | Gallbladder | 73.32 | **83.38** | 82.24 | **83.48** | 88.38 | **88.44** |
| | GI Tract | 53.26 | **79.72** | 80.53 | **86.75** | 92.02 | **93.13** |
| | Grasper | 81.52 | **90.12** | **90.30** | 88.98 | **87.00** | 86.53 |
| | Hepatic Vein | **87.78** | 85.87 | **88.03** | 86.91 | **88.74** | 87.64 |
| | L-Hook Electrocautery | **73.67** | 72.20 | **74.06** | 71.93 | 89.42 | **90.96** |
| | Liver | 55.80 | **66.08** | 63.87 | **67.55** | 72.76 | **73.40** |
| | Liver Ligament | **98.38** | 2498.25 | **98.29** | 95.66 | **98.55** | 98.52 |

ically small or low-contrast targets such as bladder, pancreas, esophagus, prostate, and spleen. Improvements are similarly pronounced in MRI, especially for cardiac structures where SAM 3 significantly outperforms SAM 2 for LV, RV, and myocardium under both single- and multi-click prompting. Multi-click prompting (1,2) reduces ambiguity for both models, yet SAM 3 retains a clear advantage in nearly all CT and MRI structures with statistically significant gains, frequently with $p < 0.001$.

For bounding-box prompts, where the spatial support is considerably less ambiguous, the performance gap narrows but does not disappear. SAM 3 continues to produce higher DSC for most CT and MRI structures, although with smaller margins. We also see some instances of SAM 2 performing slightly better than SAM 3 (e.g., MR Bladder and MR Hippocampus Posterior). Box prompts achieve the highest absolute accuracy for both models, and here the differences between the two models typically fall within a modest range ($\sim$5 DSC points) with the exception of MR Myocardium where SAM 3 beats SAM 2 by about 20 DSC points.

Ultrasound exhibits a mixed pattern. For segmentation of cardiac chambers in cine sequences (LA, LV endocardium, LV epicardium), SAM 3 achieves substantially higher DSC for click prompts, reflecting improved localization. In contrast, for the SegThy dataset (thyroid, carotid arteries, and jugular veins), both models exhibit near-total failure under click prompting, with DSCs frequently remaining in the single digits. Segmentation accuracy becomes meaningful only when bounding-box prompts are supplied; in this viable regime, SAM 2 consistently outperforms SAM 3 across the thyroid and all vascular targets.

For endoscopy (CholecSeg8K), SAM 3 shows an advantage under single-click (1,0) prompting, outperforming SAM 2 for the majority of the tissue and instrument classes. However, under multi-click (1,2) and bounding-box prompts, the results are more balanced: SAM 2 and SAM 3 each achieve higher DSC for different categories, and no model dominates across all structures. Notably, even when numerical differences are large between the two models, none of these comparisons reach statistical significance because CholecSeg8K contains only a small number of annotated videos, which limits the power of paired significance testing.

## Appendix G. Failure analysis of SegThy (3D Ultrasound) dataset

SegThy is a 3D ultrasound sweep dataset and is uniquely challenging because it combines two difficulty sources that are usually separated across modalities. In "volume-like" medical data (e.g., modalities like CT and MR), targets may first appear as tiny cross-sections near the boundary of a scan on the prompt-frame, but stable intensity statistics and sharper boundaries typically make sparse prompting tractable once the object emerges. On the other hand, in "sequence-like" medical data (e.g., CAMUS cine ultrasound), appearance is dominated by speckle and weak edges, but typically the targets are temporally continuous and occupy substantial area from the prompt-frame and across the entire sequence, providing immediate spatial support for initialization as well as continued support for propagation. SegThy inherits the prompt-frame structure ambiguity of volume-like data and the speckle-dominated, low-contrast appearance of ultrasound, so prompts must resolve the object when it is both visually ambiguous and minimally represented in pixels.

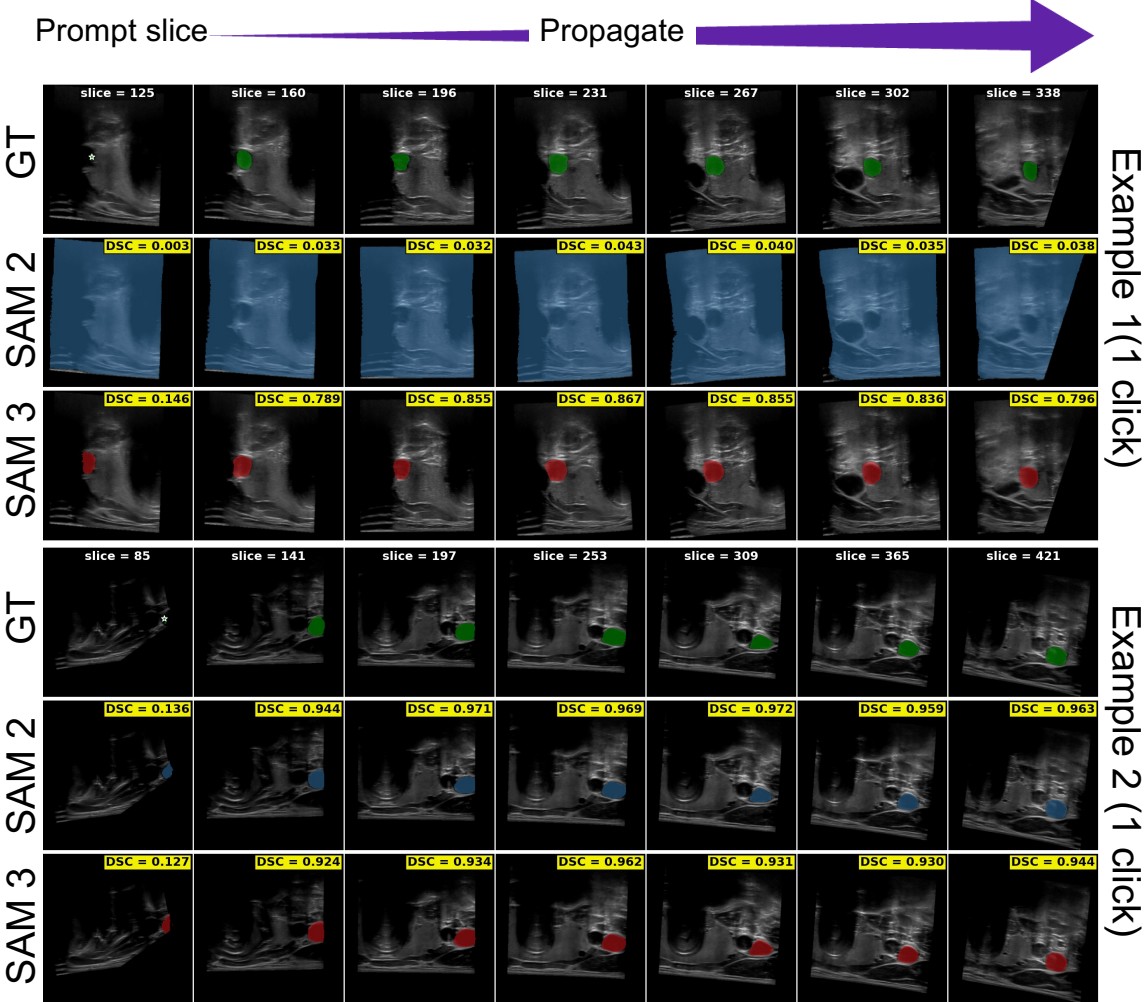

Figure 7: **SegThy click-prompt failures are driven by extreme small-target prompt-frames.** Two example cases under a single-click (1,0) prompt are shown. In Example 1, SAM 2 exhibits prompt-frame flooding and remains poorly localized across the volume (near-zero DSC), while SAM 3 is initially weak but recovers once the target occupies a larger cross-section, achieving high overlap on mid-volume slices. In Example 2, both models have low prompt-frame DSC at first appearance but track accurately on subsequent slices once sufficient pixel support emerges, illustrating that the dominant brittleness is concentrated at initialization on the prompt-frame. [Colors: GT, SAM 2, SAM 3]

In our exploration of SegThy, we found that the dominant driver of click-prompt failure is the extreme small size of the ground-truth target on the prompt-frame. Under our protocol, the prompt-frame is the first slice where the structure appears, and in SegThy this first-appearance cross-section is often only a tiny fraction of the structure's typical extent in the

same volume. Quantitatively, the ground-truth area on the prompt-frame is only $0.013\times$ the mean ground-truth area over the remaining GT-present slices in that volume, i.e., the target is typically $\sim 77\times$ smaller on the prompt-frame than on a typical slice later in the volume. In 90.5% of SegThy cases, the prompt-frame ground-truth area lies within the smallest 1% of GT-present slices in the volume. In comparison, for sequence-like datasets such as CAMUS cine ultrasound, this effect is not prevalent: the prompt-frame target size is comparable to the sequence average (median ratio $\approx 1.02$).

This extreme small-target regime impacts click prompting in two ways. First, segmentation is genuinely difficult on the prompt-frame: when the structure occupies only a few dozen pixels and is embedded in speckle-dominated texture with weak edges, sparse click prompts ((1,0) and (1,2)) often provide insufficient spatial evidence to disambiguate the target from surrounding tissue, so failures frequently begin at initialization. Second, small structures make DSC intrinsically unforgiving on the prompt-frame: even modest over-segmentation or a slight spatial offset can drive overlap close to zero when the ground truth is tiny. Consistent with this, prompt-frame predictions are frequently much larger than the ground truth in SegThy (median $\approx 581\times$ predicted-to-GT pixel count), and prompt-frame DSC under clicks is near-zero; 70.5% of cases have prompt-frame DSC < 0.01.

A counterintuitive pattern in SegThy is that full-volume DSC can be noticeably higher than prompt-frame DSC even when the initial slice is essentially missed. As the volume progresses, the target typically grows in cross-sectional area and becomes more separable from the background, so mid-to-late slices can contribute substantially more to the sequence-average DSC than the earliest first-appearance slices. Because full-volume DSC averages across hundreds of frames, later slices with larger targets can dominate the aggregate even when prompt-frame overlap is negligible. Figure 7 shows two representative SegThy volumes where the target is extremely small on the prompt-frame, leading to near-zero click initialization but delayed recovery in performance after the target grows in later slices.

SegThy is also challenging due to its long temporal extent (Table 1), which increases the opportunity for drift and error accumulation under persistent speckle and weak edges. Stronger prompts (bounding boxes or masks) largely remove the low-evidence initialization failure on the prompt frame, but propagation through hundreds of slices can still collapse after a good start. In our runs this long-horizon degradation is most visible for SAM 2, whereas SAM 3 is generally more resilient and maintains higher sequence-level DSC under strong prompts. Overall, SegThy represents a modality–geometry corner case in which click prompts are destabilized by extreme small targets on the prompt-frame, and long-horizon tracking remains difficult even with accurate initialization.

## Appendix H. Retention on all data

In the main text (Section 3.3) we focused on retention behavior for the subset of cases with good initialization ($DSC \geq 0.7$ on the prompt frame). For completeness, Figure 8 extends the same analysis to all volume–object pairs, including those with poor initialization. As expected, the distributions of normalized decay slopes broaden for both models, particularly under click prompting where failures at the first frame lead to rapid apparent decay. Under single-click (1,0) prompts, the mean slopes of SAM 2 and SAM 3 are nearly identical (approximately $-0.071$ vs. $-0.073$) and the median slopes are close to zero ($-0.006$ vs.

−0.028), reflecting that most degradation is driven by a minority of highly unstable cases. For multi-click prompting, SAM 3 retains an advantage (mean slopes −0.130 vs. −0.085; medians −0.059 vs. −0.030). The differences become more pronounced under bounding-box and mask prompts: SAM 2 exhibits substantially more negative mean slopes (about −0.262 and −0.309) than SAM 3 (about −0.152 and −0.201), and the ECDFs show that SAM 3's decay distributions are consistently shifted toward less negative values. Overall, when poor initializations are included, SAM 3 continues to forget more slowly than SAM 2 once a sufficiently strong spatial prompt is provided.

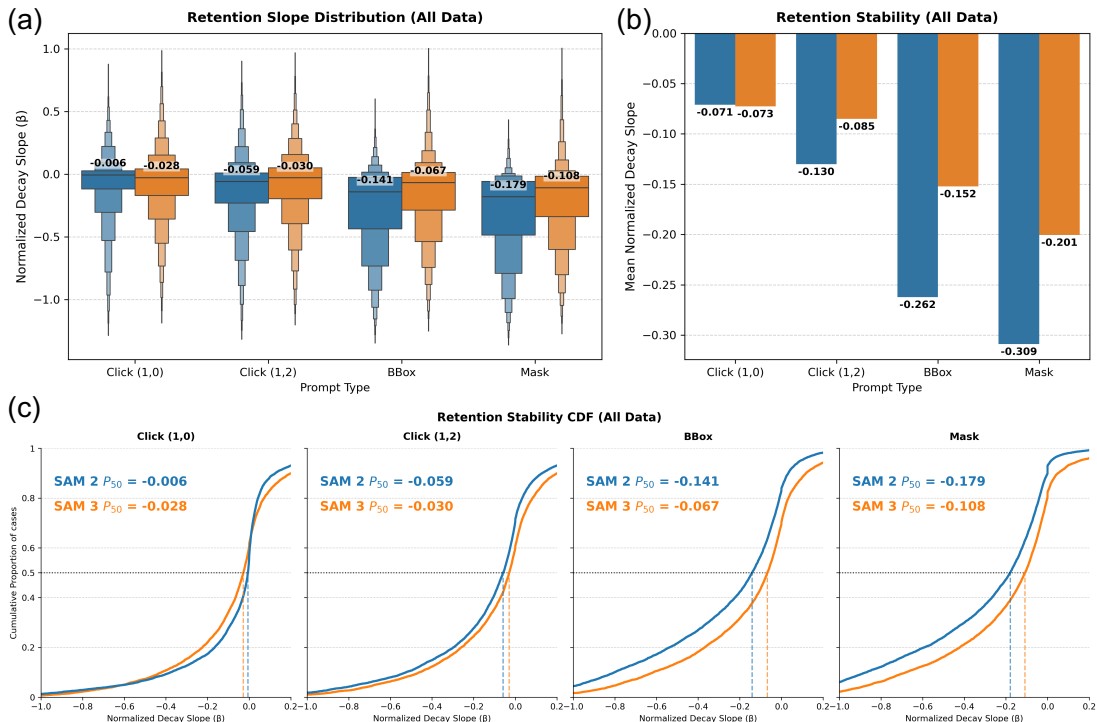

Figure 8: **Retention decay analysis on all cases.** Same layout as Figure 3, but computed over all volume–object pairs, including those with poor initialization ($DSC < 0.7$). The distributions broaden for both models, especially under click prompting, yet SAM 3 generally maintains less negative mean decay slopes for multi-click, bounding-box, and mask prompts.

