# OpenReview forum: "Comparing SAM 2 and SAM 3 for Zero-Shot Segmentation of 3D Medical Data"
_MIDL.io/2026/Validation_Papers — MIDL 2026 - Validation Papers Poster_

### Official Review · Reviewer_eiiG · 2025-12-26

**Confidence:** 4
**Preliminary Rating:** 5
**Final Rating:** 5

**Summary:**

This paper presents a systematic comparative study of SAM 2 and SAM 3 for the zero-shot segmentation of 3D medical data across four modalities: CT, MRI, ultrasound, and endoscopy. To ensure a fair comparison under a purely spatial prompting regime, the authors deliberately disabled the "concept" and language-based mechanisms of SAM 3. The study encompasses 16 public datasets covering 54 anatomical structures, pathologies, and surgical instruments. Beyond standard Dice Similarity Coefficient metrics, the research provides an in-depth quantification of three critical clinical failure modes: prompt-frame over-segmentation (flooding), temporal retention (forgetting), and over-propagation after object disappearance (stickiness). The findings conclude that while SAM 3 demonstrates superior performance in initialization and tracking complex structures, SAM 2 remains more conservative and effectively terminates tracks for compact organs under strong spatial guidance.

**Strengths:**

1. The evaluation is exceptionally thorough, spanning four imaging modalities, 16 datasets, and 54 distinct anatomical structures. This large-scale benchmark offers significant reference value for the medical imaging research community.
2. The analysis moves beyond the singular DSC metric to model specific model behaviors.
3. The authors implemented specific controls to ensure a fair comparison between the two different architectures by isolating the visual prompting encoding and propagation effects.

**Weaknesses:**

1. The categorization of anatomical structures lacks clear biological or radiological standards. Specifically, there are no defined morphological parameters to substantiate the definitions of "compact" versus "complex" structures where each model allegedly excels.
2. There are multiple technical ways to disable concept mechanisms, such as using null tokens or setting specific weights to zero. The authors should clarify the specific implementation used and justify why this method was chosen over other alternatives.
3. Since SAM 3 is fundamentally built on Vision-Language alignment, its output targets are intrinsically integrated with multimodal representations during training. Simply "masking" these architectural modules may not constitute a truly fair comparison, as the base model was optimized for a different objective. The authors should discuss whether the observed "over-propagation" (stickiness) is a byproduct of this specific architectural disabling method.

**Detailed Comments:**

None

**Justification Of Final Rating:**

I thank the authors for their detailed and convincing rebuttal. Since the authors have adequately addressed all my concerns and improved the manuscript's clarity, I maintain my rating of 5: Strong Accept and support the publication of this work.

**Justification Of The Preliminary Rating:**

This submission is highly suitable for the Validation Studies Track at MIDL. It provides a rigorous, large-scale validation of two state-of-the-art foundation models on a diverse array of 3D medical datasets. The identification of "stickiness" as a specific trade-off for SAM 3's improved retention provides critical guidance for clinical implementation. While further clarification on the architectural disabling methods and anatomical classification is needed, the depth of the failure mode analysis significantly outweighs these concerns.

**Questions To Address In The Rebuttal:**

1. Could the authors provide a deeper analysis of the near-total failure in the SegThy dataset under click prompts? Specifically, is this failure attributed to training data distribution shifts, modality-specific noise (like speckle noise in ultrasound), or fundamental zero-shot limitations of these foundation models?
2. Would using an alternative method to disable the concept modules (e.g., null tokens) yield different results regarding the "stickiness" of SAM 3?

---

### Official Review · Reviewer_zeV6 · 2025-12-30

**Confidence:** 4
**Preliminary Rating:** 4
**Final Rating:** 5

**Summary:**

This study presents a systematic comparison and evaluation of SAM2 and SAM3 across a broad range of settings. The authors benchmark the two framewroks under multiple prompting strategies and task configurations. Overall, the paper aims to clarify the practical differences between SAM2 and SAM3 rather than introducing a new method.

**Strengths:**

The study evaluates SAM2 and SAM3 on more than ten tasks, offering a comprehensive benchmark that spans diverse segmentation scenarios. And also a comparison between prompt-based and non-prompt-based settings is added. The introduction of three complementary evaluation metrics is interesting. From a methodological perspective, this multi-angle evaluation strengthens the analysis beyond standard overlap-based metrics.

**Weaknesses:**

One limitation of this study is that it focuses on only SAM2 and SAM3, without considering other variants. However, this is not a major concern given the clearly defined scope of the paper. From a broader perspective, the original SAM model offers greater practical value due to its stronger generalizability compared with fine-tuned variants. Overall, the paper does not exhibit any major weaknesses. The points raised below are mainly intended for clarification. see questions

**Detailed Comments:**

see above

**Justification Of Final Rating:**

The authors have addressed my concerns satisfactorily. While the comparative settings are not the most comprehensive, they are nonetheless meaningful and practical. I believe this study is of significant value, and I am inclined to maintain a positive evaluation. To further encourage this line of research, I am willing to raise my score. I also hope the authors will pursue future studies on SAM models in the medical field.

**Justification Of The Preliminary Rating:**

The analysis is well motivated and methodologically sound, and the conclusions are consistent with the presented results. The paper offers a useful perspective for real-world and clinical usage of SAM-based systems. Overall, I have no major concerns and find the study to be a meaningful and practical contribution.

**Questions To Address In The Rebuttal:**

1. How do the authors ensure consistency across different random click settings? To what extent does this randomness affect performance variability?

2.Could the authors provide more implementation details regarding the over-propagation setting, such as how many frames are evaluated beyond T_last ?

---

### Official Review · Reviewer_rEmz · 2026-01-10

**Confidence:** 4
**Preliminary Rating:** 3
**Final Rating:** 4

**Summary:**

The authors present the comparison between SAM 2 and SAM 3 for zero-shot 3D medical image segmentation. The experiments involve 16 public datasets encompassing 54 anatomical structures across modalities such as CT, MRI, and Ultrasound. The experiment results show that SAM 3 provides stronger initialization than SAM 2 for click prompts while under bounding box and mask, SAM 2 remains competitive.

**Strengths:**

1. Given the very recent release of SAM 3, this validation study provides immediate value to the community by assessing whether newer general-purpose architectures inherently translate to better medical domain performance.
2. The evaluation is comprehensive, covering 16 datasets and 54 distinct anatomical targets, which ensures the findings are not limited to a specific modality or organ.

**Weaknesses:**

1. Unclear Clinical Significance: While comparing SAM 2 and SAM 3 is timely, the practical utility of this comparison is unclear. Previous literature has established that the SAM family generally requires fine-tuning (e.g., MedSAM, MedSAM2) for reliable medical use.
2. While the paper summarizes performance differences, it lacks the explanation of why SAM 3 deviates from SAM 2 after the concept modules are removed. A more detailed technical discussion linking specific architectural changes (e.g., memory bank handling or transformer block updates) to the observed performance shifts is needed.
3. The study focuses exclusively on general-purpose models. Including specialized medical foundation models as baseline would better contextualize the performance gap between general and domain-specific architectures.

**Detailed Comments:**

1. The authors should clarify how 3D volumes were processed for the 2D-based encoders of the SAM models. Were they treated as independent 2D slices?
2. It is unclear if prompts were provided for every frame/slice or if the model was expected to propagate from a single initialized frame across the entire volume. Please clarify if the strategy differed between 2D and 3D datasets.
3. For visualization in Figure 6, in some slices of Example 1 and 2, there are no masks. The authors should clarify in the caption if the model failed to produce a prediction or if the target was absent. Small masks should be zoom-in to avoid confusion.

**Justification Of Final Rating:**

I thank the authors for their detailed and thoughtful rebuttal. The responses have clarified the study's scope, the technical underpinnings of the performance differences, and the practical relevance of the work. The authors have addressed my main concerns. The paper serves as a timely and comprehensive benchmark for the community. I have raised my rating to 4.

**Justification Of The Preliminary Rating:**

This paper offers a timely evaluation of the latest foundation models on a large variety of medical data. However, the scientific value of comparing two general models, one of which has its primary new features disabled, is currently not fully justified in the context of medical imaging. The SAM models are known to be limited in zero-shot medical tasks, and without a specialized medical baseline, it is difficult to determine if these incremental updates are meaningful for the field. Furthermore, clarifying the technical reasons for the performance divergence and improving the experimental detail regarding 3D input handling is necessary for a stronger recommendation.

**Questions To Address In The Rebuttal:**

1. SAM-2 and SAM-3 are general-purpose models which can not be directly used in medical imaging segmentation according to previous work (e.g. MedSAM/MedSAM-2). Please make it more clearer why directly employ these two models are practical for medical imaging instead of using domain-specific models.
2. Please include domain-specific models, e.g. foundation models like MedSAM and specialized models like nnUNet.
3. Can the authors provide a more detailed analysis of the specific architectural differences between SAM 2 and SAM 3 that lead to the observed differences in experiments?
4. Please detail the input resolution and the specific prompting protocol (iterative vs. single-shot) used for the 3D tasks.

---

### Author Rebuttal · Authors · 2026-01-25

**Rebuttal:**

We thank the reviewers for their thoughtful review. Their detailed feedback helped us improve the quality of our manuscript. We were encouraged that the submission was evaluated as a timely and useful validation study. Reviewers noted the immediate value given the recent release of SAM 3 (Reviewer rEmz), the breadth of evaluation across tasks and settings (Reviewer rEmz, zeV6, eiiG), and the inclusion of a failure-mode analysis beyond Dice (Reviewer zeV6, eiiG). Reviewers also found the methodology sound and the conclusions aligned with the results (Reviewer zeV6), and noted that the clinical failure modes provide actionable guidance (Reviewer eiiG).

Based on the reviewers' comments, we made the following major updates in the revision:
- Clarified the practical motivation and scope for using general-purpose models in interactive annotation and dataset bootstrapping (Reviewer rEmz).
- Expanded the discussion of SAM 2 vs SAM 3 design differences under visual prompting and linked them to observed failure modes (Reviewer rEmz).
- Added reproducibility details on 3D volume handling and the effective input resolution used by the inference pipelines (Reviewer rEmz).
- Specified the prompting protocol and added a click-jitter robustness analysis to quantify sensitivity to annotation variability (Reviewer rEmz, zeV6).
- Updated figures and captions to clearly mark target absence and avoid ambiguity in over-propagation interpretation (Reviewer rEmz).
- Clarified the over-propagation metric and how post-disappearance frames are counted (Reviewer zeV6).
- Removed “compact vs complex” categorization and replaced it with example-driven interpretation without implying a formal taxonomy (Reviewer eiiG).
- Clarified that SAM 3 is evaluated in Promptable Visual Segmentation (PVS) mode using the official tracker-based visual-prompt code, avoiding wording that suggests manual concept module disabling (Reviewer eiiG).
- Added a dataset-specific failure analysis for SegThy under click prompting (Reviewer eiiG).

In response to feedback, we provide individual responses below and an updated manuscript with changes highlighted in reviewer-specific color-coding: **red for Reviewer rEmz, green for Reviewer zeV6, and blue for Reviewer eiiG**.

We would again like to thank all reviewers for their time and feedback, and we hope that our revisions adequately address all concerns.

**Supporting Material:**

/attachment/b025190e2f55e9a2eb0aa9e51a7ca598d5a19613.pdf

---

### Meta-Review · Area_Chair_spNJ · 2026-02-09

**Recommendation:** Accept (Poster)
**Confidence:** 4

**Metareview:**

The paper provides a controlled, large-scale comparison of SAM 2 vs. SAM 3 for zero-shot 3D medical segmentation across multiple datasets and includes clinically meaningful failure-mode analyses. After the rebuttal, all three reviewers view the work as a timely and well-executed validation study. The conclusions are well supported, and the study should serve as a useful reference for the community.

---

### Decision · Program_Chairs · 2026-02-14

Accept (Poster)